# Neural Attention Search Linear:
# Towards Adaptive Token-Level Hybrid Attention Models

**Difan Deng** [1]  **Andreas Bentzen Winje** [1]  **Lukas Fehring** [1]  **Marius Lindauer** [1 2]

## Abstract

The quadratic computational complexity of softmax transformers has become a bottleneck in long-context scenarios. In contrast, linear attention model families provide a promising direction towards a more efficient sequential model. These linear attention models compress past $KV$ values into a single hidden state, thereby efficiently reducing complexity during both training and inference. However, their expressivity remains limited by the size of their hidden state. Previous work proposed interleaving softmax and linear attention layers to reduce computational complexity while preserving expressivity. Nevertheless, the efficiency of these models remains bottlenecked by their softmax attention layers. In this paper, we propose Neural Attention Search Linear (NAtS-L), a framework that applies both linear attention and softmax attention operations within the same layer on different tokens. NAtS-L automatically determines whether a token can be handled by a linear attention model, i.e., tokens that have only short-term impact and can be encoded into fixed-size hidden states, or require softmax attention, i.e., tokens that contain information related to long-term retrieval and need to be preserved for future queries. By searching for optimal Gated DeltaNet and softmax attention combinations across tokens, we show that NAtS-L provides a strong yet efficient token-level hybrid architecture.

## 1. Introduction

Transformers ([Vaswani et al., 2017](#)) have become the keystone of modern LLM models ([Brown et al., 2020](#); Tou-
vron et al., 2023; DeepSeek-AI, 2024b; Yang et al., 2025a). A major factor in this development is their strong ability to model long-context global information. However, self-attention operations require computing an attention map of complexity $\mathcal{O}(L^2)$. Despite that, flash attention ([Dao et al., 2022](#)) reduces the memory requirement to $\mathcal{O}(L)$ by merging the attention map directly into the attention outputs; self-attention still requires a $\mathcal{O}(L^2)$ computational cost. This computational complexity gradually becomes a bottleneck as transformers seek an even larger context length. During inference, self-attention requires a computational complexity of $\mathcal{O}(L)$ and additional $\mathcal{O}(L)$ memory complexity for caching all the KV values ([Ainslie et al., 2023](#); [DeepSeek-AI, 2024a](#)). This memory requirement poses further challenges for GPU memory systems as the model's context length increases.

Linear attention families, as an alternative to softmax attention models, have shown their efficiency in long context modeling tasks ([Katharopoulos et al., 2020](#); [Schlag et al., 2021](#); [Sun et al., 2023](#); [Dao & Gu, 2024](#); [Yang et al., 2024a](#); [Wang et al., 2025](#); [Yang et al., 2025e](#)). Instead of the non-linear softmax operations that assign non-linear Q-value-dependent weights to each V value, linear attention families apply only linear operations to compute the weighted sum for V with the QK values and therefore transform the quadratic attention operations into a linear RNN form with fixed hidden size ([Katharopoulos et al., 2020](#); [Schlag et al., 2021](#)).

However, it remains unclear whether the model can encode the entire input context into its hidden state, given its limited size. In contrast, transformer models preserve all KV context information and therefore better retain long-context information. Meanwhile, maintaining all previous information also suggests that the model might overallocate attention to irrelevant information ([Ye et al., 2025](#)).

In the ideal case, we would like to have a model that is (i) similar in its capabilities to a softmax-transformer, e.g., can preserve the past information in the context and recall this information when necessary; and (ii) similar in computational efficiency to linear-attention models (iii) able to focus only on the tokens that provide enough information for the following predictions.

---

[1]Institute of Artificial Intelligence, Leibniz University Hannover, Hannover, Germany [2]L3S Reserach Center. Correspondence to: Difan Deng <d.deng@ai.uni-hannover.de>.

*Proceedings of the $43^{rd}$ International Conference on Machine Learning*, Seoul, South Korea. PMLR 306, 2026. Copyright 2026 by the author(s).

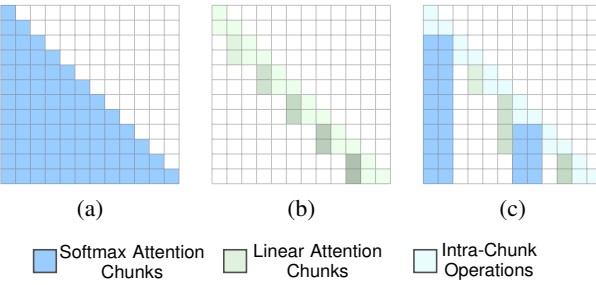

*Figure 1.* A comparison between different attention modules. In 1(a), softmax attention needs to compute each **QK**-pair. In 1(b), linear attention models perform chunk-wise computation and therefore only maintain a fixed hidden state (the color darkness indicates hidden states from different time steps). 1(c), NAtS-L automatically determines if a model belongs to softmax or linear attention and merge their output to achieve both performance and efficiency.

Several works have proposed hybrid architectures that apply different operations (Kimi-Team, 2025; MiniMax, 2025; Ren et al., 2025) in different layers to satisfy these requirements. However, these architectures have a fixed structure and may not be flexible enough to capture all required information across contexts. In this paper, we provide a different perspective: instead of having a fixed architecture where each layer is only responsible for either local or global information, we learn the importance of each token block and only apply softmax attention to tokens with high long context impact while using linear attention for the remaining tokens, as visualized in Figure 1. This allows applying softmax attention to preserve long-context related information while reducing overall computational cost with linear attention. Following the neural attention search approach (Deng & Lindauer, 2025), these optimal attention operation types can be jointly learned with model weights, resulting in a both low cost and high accuracy model.

In a nutshell, our contributions are summarized as follows:

1. We provide a unified description for non-linear softmax attention and linear attention that allows them to be processed within a single layer.

2. We propose neural attention search linear (NAtS-L), a framework that automatically determines the optimal attention operation for the input context information.

3. Experimental results show the efficiency of NAtS-L for different tasks, demonstrating that NAtS-L could achieve a better long-context modeling performance while keeping a relatively lower computational latency.

## 2. Related Work

Sofmax attention-based transformers have shown impressive abilities to model input sequence information and retrieve correlated information in long-context scenarios (Vaswani et al., 2017; Radford et al., 2019; Brown et al., 2020; Bai et al., 2024; Hsieh et al., 2024). However, the non-linear softmax attention operations require quadratic time complexity in the input context length during training and prefilling, and linear space complexity during decoding to store past KV cache values, which incurs additional computational and memory overhead in long-context scenarios.

However, in practice, transformer attention maps can be sparse and exhibit specific patterns (Zhang et al., 2023; Jiang et al., 2024; Li et al., 2024). This inspires many sparse attention variations (Child et al., 2019; Kitaev et al., 2020; Xiao et al., 2024; 2025; DeepSeek-AI, 2025; Deng & Lindauer, 2025; Yuan et al., 2025; Lu et al., 2025) that only focus on a fraction of the attention maps. Nevertheless, sparse attention models rely on a pre-defined human heuristic and ignore the correlations not covered by the sparse attention maps. Hence, imperfections in sparse attention selectors might also lead to imprecise decisions.

The computational costs of transformer models have driven researchers to seek more efficient alternatives, i.e., linear attention families (Katharopoulos et al., 2020; Peng et al., 2023; Sun et al., 2023; Dao & Gu, 2024; Yang et al., 2024b; Du et al., 2026; Guo et al., 2026). Linear attention families replace the non-linear softmax operation in the transformer with linear operations. Hence, linear transformers could either first compute the attention maps and then the weighted sum of the **V** values (the parallel form), or first compress the past **KV** values into a hidden state and then multiply the **Q** value with this hidden state (the recurrent form). The parallel form can fully utilize hardware parallelism; however, it incurs higher computational costs as the input context length grows. The recurrent form, on the other hand, processes the input data iteratively and might not make full use of the GPU computational power. Consequently, Yang et al. (2024a) propose a chunk-wise approach to combine the best of both worlds: the model first splits the input sequence into multiple chunks, the parallel form is then applied to do the computation within each chunk, while the recurrent form is applied to transfer hidden states between different chunks.

Linear attention models can encode the input sequence into a single hidden state and, therefore, enable efficient inference. However, this fixed-size hidden state might not capture all the information required in the long-context scenario. In contrast, non-linear attention models need to cache all the past KV values, which also enables them to review the entire past context and extract the corresponding information. Therefore, transformers can still often have a better performance in the long context scenario (von Oswald et al., 2025). This inspires many works on hybrid architectures that insert softmax attention layers into the linear attention layers or

heads (Dong et al., 2025; Lenz et al., 2025; Bae et al., 2025; Kimi-Team, 2025; MiniMax, 2025; Qwen-Team, 2025; Ren et al., 2025). Even so, these works still need to maintain the full attention layer, which might become a bottleneck in the long-term context scenario. Additionally, it remains unclear whether static, fixed softmax-attention/linear-attention layer ratios are optimal across different tasks and input contexts.

Another set of works proposes mixing the linear and softmax attention operations within each layer. These works include TransMamba (Li et al., 2026b), which switches from softmax attention operations to linear attention after a set of pre-defined TransPoints. However, the fixed schedule might not be flexible enough to fit different scenarios. Deltaformer (Zhong et al., 2025) uses delta rules (Widrow & Hoff, 1960) to first transform the $\mathbf{V}$ values and implicitly combine the softmax and linear attention within the same layer, but does not reduce the quadratic computational costs of softmax attention layers. Other works (McDermott et al., 2025; Zhang et al., 2025) focus on transforming softmax attention into linear attention and thus require a pre-trained network, and their performances are bounded by the softmax attention models that they are approximating. In contrast, NAtS-L adaptively determines whether to use linear or softmax attention for the current tokens. The decision process can be learned end-to-end jointly with the model weights, thereby providing a more flexible hybrid architecture without requiring a pre-trained model. We further discuss the difference between NAtS-L and other sparse models under the Appendix B.

Neural Architecture Search (Elsken et al., 2019) is a technique that searches for the optimal architecture within a search space. Previous work mainly searches for the operation within each layer and applies that operation to the entire feature map (Dong & Yang, 2019; Liu et al., 2019). Neural Attention Search (NAtS) (Deng & Lindauer, 2025) further extends this idea to search for different attention patterns within the same layer. However, as a sparse attention variation, NAtS fully ignores the correlation between the past local tokens and the current input tokens, and hence its expressibility might be degraded by this omission. In contrast, NAtS-L takes previously ignored tokens into account with linear attention, further enhancing model expressibility.

## 3. Background: Attention Operations

Both linear attention and softmax attention models utilize three matrices $\mathbf{Q}, \mathbf{K}$ and $\mathbf{V}$ to compute the attention output:

$$\mathbf{O} = f(\mathbf{Q}\mathbf{K}^\mathsf{T}, \mathbf{M})\mathbf{V}, \tag{1}$$

where $f$ is a function that transforms the attention map to enhance its expressibility and $\mathbf{M}$ is the attention mask that ensures the model's causality. This is the parallel representation form of the attention operations. If we take softmax

as $f$, we have the vanilla transformer operations[1]:

$$\mathbf{A} = \mathbf{Q}\mathbf{K}^\mathsf{T} \tag{2}$$

$$\mathbf{O} = \frac{e^A \odot \mathbf{M}}{\sum_j e^{A_{.,j}} \odot \mathbf{M}_{.,j}} \mathbf{V}. \tag{3}$$

For linear attention families, $f$ in Equation 1 is a linear function. Katharopoulos et al. (2020) show that when $f$ is a kernel function, we can first merge the KV values into one single hidden state $s_t$. This hidden state is then used to compute the final output, $\mathbf{o}_t$. This brings us the recurrent form of linear attention models:

$$s_t = g(\mathbf{k}_{0,1...t}, \mathbf{v}_{0,1,...t}) \tag{4}$$

$$\mathbf{o}_t = \mathbf{q}_t s_t, \tag{5}$$

where $g$ is a function that merges the past $KV$ values into one single hidden state $s_t$. Compared to the parallel form that requires an $\mathcal{O}(L^2 d_{head} + L d_{head}^2)$ complexity, the recurrent form reduces this complexity to $\mathcal{O}(L d_{head}^2)$, with $L$ being the context length. Given the large context length of modern LLMs, where $L \gg d_{head}$, the recurrent form requires much less computational cost.

However, in the recurrent form, the hidden states need to be updated at each time step and cannot fully utilize hardware parallelism. Hence, a more plausible hardware-friendly approach is the chunkwise parallel approach (Hua et al., 2022; Sun et al., 2023; Yang et al., 2024a). It first splits the entire sequence into multiple chunks. Then the chunkwise approach computes the hidden states and attention output in the parallel form within each chunk, and finally applies the recurrent form to transform the hidden states from one chunk to the next chunk. Assuming that we split the input sequences of length $L$ into $\frac{L}{C}$ chunks, where each chunk contains $C$ tokens. We denote $\mathbf{Q}_{[t]} \in \mathbb{R}^{C \times d_{head}}$ as the collection of all the vectors in the $t$-th chunk with $t \in [0, L/C)$ where $\mathbf{q}_{[t]}^i$ is the $i$-th vector in chunk $t$ with $i \in [1, C]$.

Hence, we compute the hidden states for each chunk and their corresponding functions as:

$$\mathbf{S}_{[t+1]} = \mathbf{S}_{[t]} + g_1(\mathbf{K}_{[t]}, \mathbf{V}_{[t]}, \mathbf{S}_{[t]}) \tag{6}$$

$$\mathbf{O}_{[t]} = \mathbf{Q}_{[t]}\mathbf{S}_{[t]}^\mathsf{T} + (\mathbf{Q}_{[t]}\mathbf{K}_{[t]}^\mathsf{T} \odot \mathbf{M}_C)g_2(\mathbf{K}_{[t]}, \mathbf{V}_{[t]}), \tag{7}$$

where $g_1$ and $g_2$ are the corresponding linear functions that update the hidden states and compute the corresponding outputs, we note that this form also applies to softmax flash-attention (Dao et al., 2022), where each time only a chunk of $\mathbf{Q}, \mathbf{K}, \mathbf{V}$ values is extracted while the output values are updated online with new chunk values.

Given that both linear attention and softmax attention require a chunk-wise form to compute the output, the model

---

[1] For the sake of simplicity, we omit the scaling factor $\frac{1}{\sqrt{d_{attn}}}$.

could learn to determine the optimal attention operation type within each chunk that best fits the current input.

## 4. Token Level Hybrid Attention Architecture

We now construct a search space that contains both linear and non-linear attention operations. We first show how to combine different attention operations with the sampled token types in our search space. We then demonstrate how the model learns the optimal operation combinations by gradient information. Finally, we describe the overall NAtS-L architectures.

### 4.1. Chunk-wise Hybrid Attention

Moving beyond chunk-wise linear attention, NAtS-L assigns each chunk of tokens to either utilize softmax or linear attention. Given an input sequence $\mathbf{X} \in \mathbb{R}^{L \times d}$, we first split it into chunks with each chunk of $\mathbf{X}_{[t]} \in \mathbb{R}^{C \times d}$. Following NAtS (Deng & Lindauer, 2025) and MoE (Shazeer et al., 2017; Fedus et al., 2023; DeepSeek-AI, 2024b; Du et al., 2026), we apply an Attention Score Layer that maps the input feature map within each chunk into scores for each operation. This Attention Score Layer is a mean-pooling layer followed by another linear layer (Yuan et al., 2025) that maps an entire chunk to a set of score values without introducing too much computational overhead,

$$score_t = \mathbf{W}^{score} Mean(\mathbf{X}_{[t]}) \in \mathbb{R}^{\frac{L}{C} \times N_{\text{opts}}}. \quad (8)$$

The attention type with the highest score for each chunk is then assigned to that chunk.

$$opt_t \in \arg\max\ softmax(score_t). \quad (9)$$

Assuming that we group the input chunks into two parts: chunks belonging to the linear attention families $t_{la} = \{t | \mathbf{X}_t \text{ is linear chunk}\}$ and chunks belonging to the non-linear operations $t_{nla} = \{t | \mathbf{X}_t \text{ is non-linear chunk}\}$. We then construct a column-wise learnable attention mask for each of the corresponding attention maps and rewrite Equation 1 and 3 as:

$$\mathbf{O}_{la} = f(\mathbf{Q}\mathbf{K}_{la}^{\mathsf{T}} \odot \mathbf{M}^{la})\mathbf{V}_{la} \quad (10)$$

$$\mathbf{O}_{nla} = \frac{e^A \odot \mathbf{M}^{nla}}{\sum_j e^{A_{\cdot,j}} \odot \mathbf{M}^{nla}_{\cdot,j}} \mathbf{V}_{nla}, \quad (11)$$

where each column of the $\mathbf{M}$ is filled by its corresponding attention types:

$$\mathbf{M}^{nla}_{i,j} = \begin{cases} 1, & \text{if } j \in t_{nla}\ \&\ i \geq j \\ 0, & \text{if } j \in t_{la}\ \&\ i < j \end{cases} \quad (12)$$

$$\mathbf{M}^{la}_{i,j} = \begin{cases} 1, & \text{if } j \in t_{la}\ \&\ i \geq j \\ 0, & \text{if } j \in t_{nla}\ \&\ i < j \end{cases}. \quad (13)$$

Once we compute the outputs for the linear and non-linear attention, we can sum them together as the output of NAtS-L. We detailed this process in Section 4.3.

Hence, we skip $\mathbf{KV}$ values that do not belong to the corresponding attention types to accelerate the forward and backward process: for softmax attention modules, we only load the tokens belonging to the non-linear chunks within each flash-attention iteration (Dao et al., 2022). For linear attention modules, we only update the chunk-wise hidden states when the corresponding chunk belongs to linear attention families:

$$\mathbf{S}_{[t+1]} = \begin{cases} \mathbf{S}_{[t]} + g(\mathbf{K}_{[t]}, \mathbf{V}_{[t]}) & \text{if } t \in t_{la} \\ \mathbf{S}_{[t]} & \text{if } t \notin t_{la} \end{cases}. \quad (14)$$

The Attention Score Layer provides the score only after all hidden states in the chunk have been observed. Therefore, we ask all the operations in our search space to compute the inner-chunk correlation as the output for each time step.

Finally, we merge the attention output from the two models. As a result, the overall computational complexity decreases to $\mathcal{O}(L_{nla}L + L_{la})$, with input sequence length $L$, the number of tokens for non-linear attention $L_{nla}$, and the number of tokens for linear attention operations $L_{la}$, respectively.

Figure 1(c) shows an exemplary process of computing the attention outputs with NAtS-L, with a chunk size of 2. Chunks $1, 4$ are classified as softmax attention chunks while chunks $2, 3, 5$ are modeled with linear attention. Hence, the softmax attention operations involve the full $\mathbf{Q}$ matrix and the $\mathbf{KV}$ values from chunks 1 and 4. Additionally, we utilize linear attention three times for chunks 2, 3, and 6.

Since we can determine the operations applied to each chunk after observing all of the chunk's tokens, we first apply both linear and softmax attentions to construct intra-chunk correlations. During per-token generation, we maintain the hidden states of both linear attentions and softmax attentions, i.e., the $KV$ caches. Once we arrive at a new chunk, we pseudo-update both hidden states and, after observing the entire chunk, reroll the hidden states of the operations that are not selected as the target operation. More specifically, we remove the new $KV$ cache values from chunks that are classified as linear attentions and reset the hidden states to the start of the chunk for softmax attention chunks. Thus, compared to full-attention models, we efficiently reduce overall memory and computational overheads, thereby accelerating the inference process.

### 4.2. Optimizing for the Optimal Operation Combinations

We now compute the gradients for the Attention Score Layer. To compute the gradient for sotmax attention mask $\mathbf{M}^{nla}$,

we first define $P_{i,j} := \frac{e^{\mathbf{A}_{i,j}}}{\sum_j e^{A_{.,j} \odot \mathbf{M}_{.,j}^{nla}}}$, the gradient for the Attention Score Layer is then computed by the column-wise sum of the masks' gradient values (Deng & Lindauer, 2025; Dao et al., 2022):

$$\mathrm{d}\mathbf{M}_{i,j}^{nla} = P_{i,j}(\mathrm{d}P_{i,j} - \mathrm{d}o_i^{\mathsf{T}} o_i) \tag{15}$$

$$\mathrm{d}score_t^{nla} = \sum_{\substack{0 \le i \le T \\ tC \le j \le (tC+C)}} \mathrm{d}\mathbf{M}_{i,j}^{nla}. \tag{16}$$

Since $P_{i,j}$ and $dP_{i,j}$ are both the intermediate variables required by attention computation, we seamlessly integrate Equation 15 into the flash-attention (Dao et al., 2022) forward and backward processes.

Since $\mathbf{O}_{la}$ is the linear combination of $\mathbf{S}$ and $\mathbf{M}^{la}$, we directly compute the gradient for the linear attention scores with

$$\mathrm{d}score_t^{la} = \sum (\mathrm{d}\mathbf{S}_{[t]} \cdot \mathbf{S}_{[t]}). \tag{17}$$

The model can automatically learn the optimal attention type for each chunk with the gradient information. This gradient information is computed jointly with the gradient for the $\mathbf{Q}, \mathbf{K}, \mathbf{V}$ values. The gradient for $\mathbf{k}, \mathbf{v}$ becomes $0$ for a given operation if a chunk containing these tokens is inactive for the corresponding operation. However, computing the gradient values $\mathrm{d}score$ for all the activate and inactivate operations requires a computational complexity of $\mathcal{O}(L^2 + L_{la})$ since we need to iterate over the inactivate operations. To reduce this computational cost, we do not compute the score gradients for inactive chunks and set $\mathrm{d}score = 0$ if $\mathbf{M}_t = 0$. We therefore incur the same computational cost for both the backward and forward passes.

### 4.3. Hybrid Architecture as Token Mixer

Here, we show how to search for the optimal hybrid operations within a search space that contains softmax attention and Gated DeltaNet (GDN) (Yang et al., 2025e), an improved version of DeltaNet (Schlag et al., 2021). DeltaNets apply delta rules (Widrow & Hoff, 1960) to update their hidden states:

$$\mathbf{S}_t = \mathbf{S}_{t-1}(\mathbf{I} - \beta_t \mathbf{k}_t \mathbf{k}_t^{\mathsf{T}}) + \beta_t \mathbf{v}_t \mathbf{k}_t^{\mathsf{T}}. \tag{18}$$

Yang et al. (2024b) showed that Equation 18 can be written in the chunk-wise parallel form and proposed an efficient approach to train DeltaNet with long input length.

GDN further aplies a decay term $\alpha_t$ (Dao & Gu, 2024) to adaptively manage the historical memory:

$$\mathbf{S}_t = \mathbf{S}_{t-1}(\alpha_t(\mathbf{I} - \beta_t \mathbf{k}_t \mathbf{k}_t^{\mathsf{T}})) + \beta_t \mathbf{v}_t \mathbf{k}_t^{\mathsf{T}}. \tag{19}$$

In practice, we always apply the decay $\alpha_t$ towards the linear attention's hidden states, even for the non-linear attention

chunks. This enforces linear attention models to focus on the most recent information and to forget information farther in the past:

$$\mathbf{S}_{[t+1]} = \prod_{i \in [t]} \alpha_i \mathbf{S}_{[t]} \quad \text{if } t \notin t_{la}. \tag{20}$$

To fully utilize the hardware parallelism, avoid computational fragmentation, and accelerate the training and inference process, we set the NAtS chunk size to be no smaller than the GDN chunk size. In the meantime, an overly large chunk size might not flexibly adapt to the corresponding input context. Hence, we set NAtS-L chunk size identical to the GDN chunk size. Additionally, the linear attention families typically require a larger number of hidden states to contain enough historical information and thereby require fewer heads. In contrast, the transformer families might prefer more heads to construct different correlations among heads. Hence, similar to the GQA model (Ainslie et al., 2023), we group multiple transformer heads that share the same set of attention types and the corresponding masks. We set the number of NAtS-L heads equal to the number of linear attention families, i.e., each linear attention head receives its own mask.

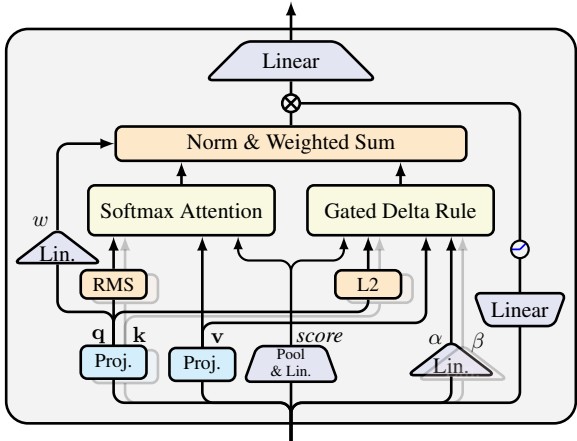

*Figure 2.* The NAtS-L Architecture. The projection layers for $\mathbf{q}, \mathbf{k}, \mathbf{v}$ are a linear layer followed by a short conv and SiLU activation function. The scores for each operation are computed by a mean pooling layer followed by a linear layer (Equation 8). The model then selects the operations with the highest score for each chunk and feeds them to different attention operations. The outputs from each attention model are then normed and weighted summed (Equation 21), where the weights $w$ for each attention output are mapped from the $\mathbf{q}$ matrix.

However, though Equations 10 and 11 both compute the weighted sum for $\mathbf{V}$, they are still represented in different forms: the softmax attention families normalize the input weights to be positive and sum to 1, and the output from the linear attention families is influenced by the $\mathbf{QK}$ norm

values, which might not match the scale of the softmax attention operations. Hence, we first normalize each output individually with Root Mean Square (RMS) normalization (Zhang & Sennrich, 2019) and then sum them together with time-dependent head-wise weights:

$$\mathbf{O}_t = w_t^{nla} Norm(\mathbf{O}_t^{nla}) + w_t^{la} Norm(\mathbf{O}_t^{la}). \quad (21)$$

The weights $\mathbf{w}^{nla} \in \mathbb{R}^{L \times H}$ and $\mathbf{w}^{la} \in \mathbb{R}^{L \times H}$ assign scalar weights to each head. Since the attention output values are determined by how well each $\mathbf{q}$ matches with the other $\mathbf{k}, \mathbf{v}$ values, we compute these weights with a linear layer from the attention $\mathbf{q}$ values, i.e., the output from the $\mathbf{q}$-projection layers. We empirically apply softmax over the output weights to ensure that the two weights are properly normalized:

$$\begin{bmatrix} \mathbf{w}^{nla} \\ \mathbf{w}^{la} \end{bmatrix} = softmax(\begin{bmatrix} \mathbf{W}^{nla} \\ \mathbf{W}^{la} \end{bmatrix} \mathbf{Q}). \quad (22)$$

The overall NAtS-L architecture is shown in Figure 2. This design decision follows the design from the previous linear attention blocks (Sun et al., 2023; Dao & Gu, 2024; Yang et al., 2025e; Kimi-Team, 2025): $\mathbf{Q}, \mathbf{K}, \mathbf{V}$ are generated by feeding the input feature maps with a linear layer followed by a short depthwise convolutional layer and a SiLU activation function. We apply L2-Normalization for the linear attention $\mathbf{QK}$ values and RMS norm for the softmax attention $\mathbf{QK}$. Finally, we apply a gating layer to the attention module output and feed the result to the output linear layer. Hence, the additional parameters that NAtS-L brings to the vanilla linear attention models come from the two projection layers that compute the *score* and the output weights for the two attention models, negligible compared to the number of parameters in the model.

Both attention families share the same sets of $\mathbf{Q}, \mathbf{K}, \mathbf{V}$ to reduce the model's parameter numbers. Since Equation 10 and 11 take different subsets of $\mathbf{K}$ and $\mathbf{V}$, and are independent from each other, NAtS-L could also be considered as a special case for Context Parallel (Liu et al., 2024; Yang et al., 2025b) with heterogeneous operations. Additionally, we could also apply different weights for non-linear and linear attention modules, resulting in a mixture-of-expert (MoE) model for attention operation (Piekos et al., 2025; Du et al., 2026). However, for the sake of fair comparison and to keep the number of model parameters consistent across different models, we will focus on the case where all the $\mathbf{Q}, \mathbf{K}, \mathbf{V}$ are shared across different operations in this paper. Additionally, unlike other approaches that introduce auxiliary losses or other mechanisms to control the load between different experts (or operations) (Fedus et al., 2023; DeepSeek-AI, 2024a), we do not assign any constraints to the softmax attention-linear attention ratio. The distribu-

tions for softmax and linear attention are determined solely by the language modeling loss.

## 5. Experiments

We perform academic-scale pre-training tasks on the Fineweb-Edu dataset (Lozhkov et al., 2024) with (i) 380M parameters for 15B tokens.[2] (ii) 800M parameters for 50B tokens. Models in both setups are trained with a context length of 4096. We compare NAtS-L with the following baselines: Gated Delta Net (GDN, 21 layers, 793M) (Yang et al., 2025e), Mamba2 (48 layers, 801M) (Dao & Gu, 2024), softmax attention transformers (24 layers, 778M) (Vaswani et al., 2017) with Rope positional encoding (Su et al., 2024). Additionally, we compare NAtS-L with the layer-wise GDN and transformer hybrid model (GDN Hybrid), where the ratio between the linear and non-linear model is 3:1 (5 transformer layers and 17 GDN layers, 802M) (Kimi-Team, 2025). We use the implementation of all backbones from the flash-linear-attention library (Yang & Zhang, 2024) and train the model with the flame package (Zhang & Yang, 2025). For the NAtS-L backbone, we test two variations, the first variation only contains NAtS-L layers (NAtS-L, 21 layers, 794M) with Rope for the softmax attention opeartions, while for the second variation, we insert NAtS-L into the GDN layers, similar to the GDN-Hybrid architecture (NAtS-L Hybrid, 6 NAtS-L layers and 15 GDN layers,793M); however, we replace the GDN attention operations with NAtS-L operations and no positional encoding is applied to the softmax attention related values. We set the chunk size for all NAtS-L operations to 64, matching the GDN implementation.

The detailed model hyperparameters are listed in the appendix A.1. All experiments are run on a cluster, where each node is equipped with 4 NVIDIA H100 PCIe GPUs with 80 GB of VRAM. We train the models with 4 or 8 GPUs, depending on the architecture scales [3].

### 5.1. Language Modelling

We first evaluate the models on several zero-shot commonsense reasoning benchmarks. Here we consider LAMBADA (LMB.) (Paperno et al., 2016), PIQA (Bisk et al., 2020), Hellaswag (Hella.) (Zellers et al., 2019), Wino-Grande (Wino.) (Sakaguchi et al., 2020), OpenbookQA (OQA.) (Mihaylov et al., 2018), ARC-easy and ARC-challenge (Clark et al., 2018). These benchmarks focus on short-context tasks and require the model's intrinsic knowledge; therefore, performance is mainly influenced by the

---

[2]The results of these smaller models are presented in the appendix C.1.

[3]The code to reproduce our result can be found under https://github.com/automl/NeuralAttentionSearchLinear

model parameter sizes. As shown in the left part of Table 1, all models achieve similar performance, with NAtS-L Hybrid and GDN Hybrid performing slightly better than the others.

Next, we evaluate models on real-world retrieval tasks (Arora et al., 2024a;b). These tasks require the model to extract key-related values from the input context and, as a result, pose further challenges for assessing whether the model can preserve important context information in its hidden states to respond to input keys. We truncate all input contexts to 4096, i.e., the same length as the training context for all models. The result is illustrated in the right part of Table 1. Overall, NAtS-L Hybrid and NAtS-L achieve the best average scores. Specifically, NAtS-L Hybrid outperforms the other models on five out of six benchmarks while NAtS-L achieves the highest accuracy on the DROP benchmark. We note that models with only one architecture type fail on specific tasks, e.g., GDN and Mamba2 on FDA, and Transformer on SQD, while both NAtS-L variations perform equally well on different tasks, further showing their robustness in context modeling ability.

We further test the length extrapolation ability of different models by evaluating their perplexity on different tasks with an input context length of $65\,536$: PG19 (Rae et al., 2020), CodeParrot, and NarrativeQA (Kociský et al., 2018). The PG19 and NarrativeQA datasets have an average per-sample length of more than $60k$ tokens, which mainly requires long-term correlation. For CodeParrot, the average context length is much smaller; hence, we need to concatenate multiple samples to reach the desired context length. This adapted benchmark still focuses more on short-term correlations. The result is demonstrated in Figure 3. Overall, GDN Hybrid achieves slightly lower in-context perplexity but fails when the input context length exceeds the training context length. At the same time, NAtS-L and NAtS-L Hybrid maintain most of its perplexity even beyond the training context length, indicating the importance of mixing softmax and linear attention within each input sequence. Additionally, NAtS-L Hybrid achieves the lowest per-token perplexity among the approaches, indicating that attaching softmax attention under the sequence level also helps with long-context modeling tasks. Interestingly, on the NarratieQA dataset, NAtS-L Hybrid consistently decreases its perplexity as the input context grows until it reaches the same level as on PG19 (roughly $15.0$), which might indicate that the NarrativeQA dataset would require even longer context information to be correctly modeled. Hence, models with better long-context modeling ability could continually improve their performance as input context lenght increase.

We now evaluate all models on the long-context benchmarks RULER (Hsieh et al., 2024) and LongBench (Bai et al., 2024). For the RULER benchmark, we test with input context lengths of $4k$, $8k$, and $16k$ on the retrieval tasks. Since all the models are trained only with a context length of 4096, this also tests whether they can extrapolate beyond that length. For the LongBench task, we evaluate them on the LongBench-e subsets and report results across different context ranges.

Table 2 shows the results on the RULER benchmark. NAtS-L Hybrid archives the best performance for the input context length $4k$, $8k$, and $16k$. When the input context length grows to $16k$, NAtS-L Hybrid achieves a performance comparable to the linear attention variation with input context length of $4k$, showing the importance of preserving softmax attention tokens in the retrieval tasks. The detailed results for each task can be found in the appendix C.3. Table 3 illustrates the results on Longbench with input context length until $4k$. NAtS-L Hybrid achieves the best performance on 5 out of 11 benchmarks. We put the results for the other context length in the appendix C.4

Figure 4 illustrates the pre-filling and decoding time for different models with different input context lengths. Under the longest input sequence, NAtS-L Hybrid is only 1.66x slower than the GDN model for the pre-filling and achieves a 5.4x speedup compared to the transformer model, respectively. For decoding time, NAtS-L Hybrid achieves a 2.3x speed up compared to the transformer model for the input context length of $128k$.

### 5.2. Token Type Distribution

We now study how different token types are distributed within the model. We collect the number of softmax attention tokens within each layer of different tokens. As shown in Figure 5, for the text datasets PG19 and NarrativeQA, the softmax attention token distributions are closer to each other than the distributions on the code benchmark. e.g., both NarrativeQA and PG19 require a higher number of softmax attention tokens for head 6, layer 4, which is much less used by the CodeParrot dataset. This shows that NAtS-L could adapt its token distributions to different input contexts. Despite that, we can see an overall trend: several heads contain only linear attention, while the others are a mix of linear and softmax attention. Additionally, although some heads only contain linear attention operations, no head contains pure softmax attention operations (the head 1 in layer 4 is close, but it could still generate linear attention tokens). This might indicate that a softmax attention towards the entire sequence might not always be the optimal solution for sequence modeling tasks.

Figure 6 illustrates the fraction of the softmax attention-linear attention within each chunk on the PG19 dataset (Rae et al., 2020). Overall, most heads apply only one attention

*Table 1.* (Evaluation results on language modeling and zero-shot common sense reasoning tasks(left) and retrieval tasks with input truncated to 4096 (right). The performance is evaluated with the model size of 800M. Best results are bold; second-best are underlined.

| Model | Wiki. ppl↓ | LMB. ppl↓ | LMB. acc↑ | PIQA acc↑ | Hella. acc↑ | Wino. acc↑ | OQA. acc↑ | ARC-c acc↑ | ARC-e acc↑ | Avg. | SWDE acc↑ | SQD acc↑ | FDA acc↑ | TQA acc↑ | NQ acc↑ | DROP acc↑ | Avg. |
|---|---|---|---|---|---|---|---|---|---|---|---|---|---|---|---|---|---|
| GDN | 19.54 | 16.77 | 42.15 | 71.06 | 50.72 | 55.01 | 36.60 | 33.87 | 67.76 | 51.02 | 34.47 | 35.05 | 22.69 | 56.87 | 17.10 | 21.13 | 31.22 |
| Mamba2 | 19.28 | 17.54 | 41.01 | **71.60** | **51.73** | 54.54 | 37.60 | **35.24** | 67.93 | 51.38 | 32.94 | 35.76 | 17.70 | 55.98 | 17.64 | 19.65 | 29.94 |
| Transformer | 19.27 | 17.89 | 41.88 | 70.78 | 50.58 | 54.22 | 37.80 | 33.53 | 66.67 | 50.78 | 58.42 | 3.25 | 43.47 | 56.69 | 18.28 | 21.85 | 33.66 |
| GDN Hybrid | 18.83 | 16.41 | 42.56 | 70.95 | 51.60 | **57.93** | 35.40 | 34.04 | **68.18** | **51.52** | 54.64 | 39.54 | 31.13 | 57.11 | 19.64 | 21.75 | 37.30 |
| NAtS-L | 19.17 | 16.17 | 42.34 | 70.24 | 50.27 | 54.54 | **39.60** | 32.94 | 65.95 | 50.84 | 53.02 | 37.87 | 51.54 | 55.75 | 17.42 | **22.90** | 39.75 |
| NAtS-L Hybrid | **18.52** | **15.16** | **44.30** | 70.67 | 51.10 | 55.96 | 38.60 | 32.51 | 66.75 | 51.41 | **62.38** | **40.01** | **67.70** | **57.64** | **21.86** | 20.84 | **45.07** |

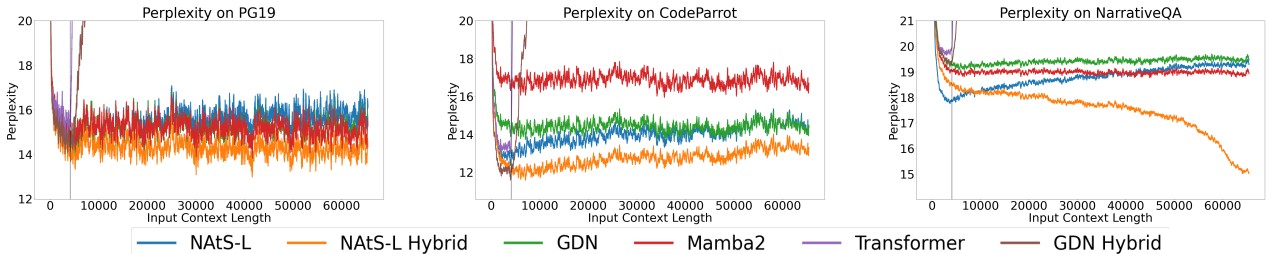

*Figure 3.* Per-token perplexity on different datasets, all the models are trained with 4096 tokens (the black vertical line)

*Table 2.* Mean Scores for RULER benchmarks with different input context length. Higher is better.

| Model | 4k | 8k | 16k |
|---|---|---|---|
| GDN | 0.25 | 0.14 | 0.04 |
| Mamba2 | 0.18 | 0.09 | 0.04 |
| Transformer | 0.45 | 0.00 | 0.00 |
| GDN Hybrid | 0.47 | 0.02 | 0.00 |
| NAtS-L | 0.39 | 0.13 | 0.05 |
| NAtS-L Hybrid | **0.49** | **0.32** | **0.21** |

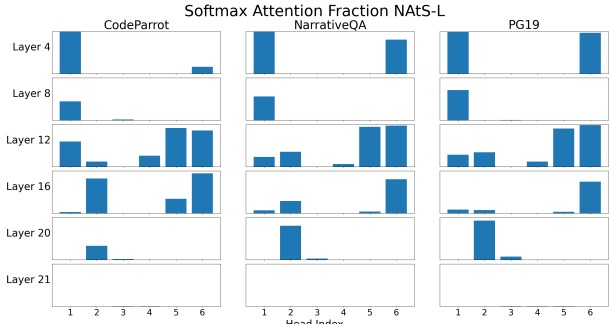

*Figure 5.* Fraction of softmax attention tokens within each layer for NAtS-L Hybrid in different tasks.

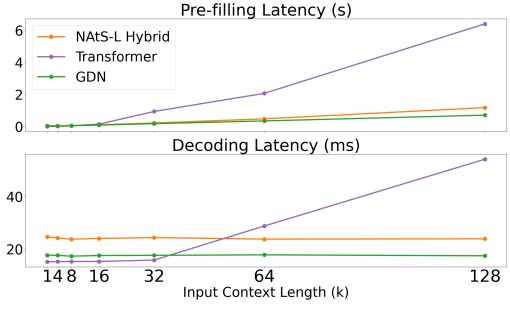

*Figure 4.* Inference Latency with different input context length

operation across all time steps. However, some heads, i.e., the first heads in the 8th layer, gradually increase the fraction of softmax attention operations as the number of observed tokens increases. This further highlights the importance of assigning each operation individually to various tokens. Further results can be found under Appendix C.5

## 5.3. Ablation Study

In this section, we study the design decision made in Section 4.3. We train NAtS-L Hybrid with different settings and evaluate them on the information retrieval benchmarks. We consider the following variations: (i) In Section 4.1, we state that both operations are involved in the inner-chunk computation. Here we study two variations: NAtS-L that only apply inner-chunk correlation with softmax attention (*SAttn Out*) and GDN (*GDN Out*). (ii) In Equation 20, we apply the decay to the hidden states even for the softmax chunks, and we now study if keeping the hidden states unchanged provides better results (*w/o LAttn Decay*). (iii) Equation 21 shows that the outputs are first normalized and then summed with the corresponding weights. We study two variations here: the first one sums the two attention outputs and only applies a single normalization layer for the attention output (*w/o Attn Norm*), the second one does not use the weights in Equation 21 (*w/o Attn Weights*).

*Table 3.* Evaluation results for LongBench benchmarks with input context length below $4k$: 2WikiMultihopQA (2WM), HotpotQA (HQA), MultiFieldQA-En (MFQ), Qasper (QQA), MultiNews (MN), GovReport (GOV), TREC (TRC), TriviaQA (TQA), SAMSum (SSM), LCC (LCC), RepoBench-P (RBP).

| Model | 2WM | HQA | MFQ | QQA | MN | GOV | TRC | TQA | SSM | LCC | RBP |
|---|---|---|---|---|---|---|---|---|---|---|---|
| GDN | 11.72 | 7.38 | 14.43 | 4.61 | 11.50 | 10.18 | 24.00 | 18.46 | 24.30 | 17.22 | 17.85 |
| Mamba2 | **12.04** | 7.32 | 13.36 | **4.79** | 9.35 | 11.15 | 6.00 | 21.72 | 13.59 | **20.68** | 20.75 |
| Transformer | 3.74 | 3.48 | 9.39 | 2.74 | 13.30 | 12.41 | 18.00 | 12.48 | 16.36 | 12.05 | 10.39 |
| GDN Hybrid | 11.87 | 7.02 | 13.17 | 4.28 | **13.90** | 12.54 | **41.00** | 20.10 | 16.21 | 20.11 | 16.43 |
| NAtS-L | 10.94 | 6.93 | 14.81 | 4.48 | 9.59 | 11.09 | 22.00 | 23.96 | 19.36 | 18.11 | **20.82** |
| NAtS-L Hybrid | 11.31 | **7.45** | **15.12** | 2.40 | 10.17 | **17.26** | 17.00 | **29.79** | **24.98** | 9.43 | 15.21 |

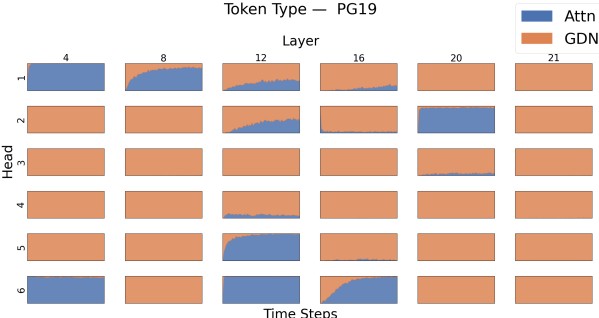

*Figure 6.* Time-Wise Token distributions for PG19 dataset.

(iv) Equation 22 normalizes the output weights of different operations with softmax. In this ablation, we study two other variations to replace the softmax activation function: first, using a sigmoid activation function (*Sigmoid Weights* and second, not using any activation function (*Linear Weights*)) (v) In Figure 2, we show that the attention output weights are mapped from **q** values; this variant instead computes the output weights from the input feature map **X** (*Weighs From* **X**). (vi) We set the NAtS-L chunk size as 64 for all variations. Here, we increase the chunk size to 128 and evaluate how it influences model performance.

*Table 4.* Ablation Study on the design decisions in NAtS-L.

| Model | SWDE | SQD | FDA | TQA | NQ | DROP | Avg. |
|---|---|---|---|---|---|---|---|
| NAtS-L Hybrid | 44.37 | 32.91 | **42.20** | 46.15 | **16.53** | 18.78 | **33.49** |
| Attn Out | 40.95 | 30.80 | 40.56 | 44.08 | 14.06 | 14.76 | 30.87 |
| GDN Out | 35.10 | 28.05 | 27.13 | 45.08 | 10.77 | 15.72 | 26.98 |
| w/o LAttn Decay | **46.26** | 32.17 | 34.75 | 45.38 | 15.33 | 19.55 | 32.24 |
| w/o Attn Norm | 36.18 | 32.47 | 30.76 | 46.80 | 11.72 | 19.60 | 29.59 |
| w/o Attn Weights | 42.39 | **33.85** | 34.03 | 47.69 | 16.19 | **19.89** | 32.34 |
| Sigmoid Weights | 44.91 | 28.65 | 31.58 | 44.37 | 14.76 | 15.81 | 30.02 |
| Linear Weights | 41.76 | 33.04 | 28.86 | **47.75** | 14.41 | 17.59 | 30.57 |
| Weights From **X** | 32.49 | 29.83 | 27.68 | 45.85 | 12.32 | 17.68 | 27.64 |
| NAtS-L Chunk 128 | 30.78 | 28.42 | 32.85 | 45.32 | 10.86 | 18.88 | 27.85 |

The result is shown in Table 4; overall, our current decision achieves the best score among all our variations.

## 6. Conclusion and Future Work

We introduced NAtS-L, a token-level hybrid attention model that adaptively determines the optimal attention operation for each input token. We show that the hybrid attention operations provide a better long-context modeling ability while keeping an overall small computational latency.

The current NAtS-L contains only two operations: softmax attention and GDN, as these two operations are also widely utilized in other hybrid models (Kimi-Team, 2025; Qwen-Team, 2025). However, Fu et al. (2025) shows that interleaving between different linear attention models could yield a better performance. Hence, a potential future direction is to expand our search space to provide an even stronger hybrid architecture. Additionally, we do not assign any auxiliary losses regarding the desired amount of softmax or linear attention tokens. Regularizing the amount of overall softmax attention or linear attention tokens could be an interesting future direction that provides a better efficiency-performance trade-off (Deng & Lindauer, 2025).

## Acknowledgements

Difan Deng was supported by the Federal Ministry of Education and Research (BMBF) under the project AI service center KISSKI (grantno.01IS22093C). Andreas Bentzen Winje was supported by the German Federal Ministry for the Environment, Climate Action, Nature Conservation and Nuclear Safety (BMUKN) (GreenAutoML4FAS project no. 67KI32007A). Lukas Fehring and Marius Lindauer acknowledge funding by the European Union (ERC, "ixAutoML", grant no.101041029). Views and opinions expressed are however those of the author(s) only and do not necessarily reflect those of the European Union or the European Research Council Executive Agency. Neither the European Union nor the granting authority can be held responsible for them.

The authors gratefully acknowledge the computing time provided to them on the high-performance computers Noctua2 at the NHR Center PC2 under the project hpc-prf-intexml. These are funded by the Federal Ministry of

Education and Research and the state governments participating on the basis of the resolutions of the GWK for the national high performance computing at universities (www.nhr-verein.de/unsere-partner).

## Impact Statement

This paper proposes a dynamic hybrid attention framework to improve both model efficiency and long context modeling ability. The improved efficiency enables a more powerful model on the resource-constrained device, while the retrieval ability may help it retrieve the required information more effectively, potentially reducing hallucination. However, whether the new attention mechanism can reduce model bias remains unexplored and warrants further study.

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

## A. Experiment Details

### A.1. Model Hyperparameters

Here, we list all the detailed model architectures and hyperparameters applied for training the model. All the models are trained with AdamW (Loshchilov & Hutter, 2019) with a peak learning rate of $3e - 4$. Both models are trained with a batch size of 0.5M tokens with gradient accumulation, where the 380M models are trained with 15B tokens, and the 800M models are trained with 50B tokens. We use a cosine annealing learning rate schedule with warmup of 0.5B (for 15B training tokens) and 1B (for 50B training tokens). The initial and ending learning rates are set as $3e - 5$.

Models with different parameter scales have the same number of layers, but differ in the network width. All the GDN and NAtS-L have 21 layers, while the transformer and mamba2 models have 24 and 48 layers, respectively. For the hybrid model, the linear and non-linear ratios are set to 3:1. Since the transformer blocks have fewer parameters per layer, we use 22 layers for the GDN Hybrid blocks with 5 transformer layers and 17 GDN layers. Models that require short convolutional operation has a kernel size of 4. Models with 380M parameters have a hidden state of 1024. This value increases to 1536 for models with 800M parameters. All the GDN layers have 6 heads across different parameter scales. However, for the other operations, the number of heads scales with the number of parameters: mamba2 has 32 and 48 heads, while the transformer has 16 and 24 heads, respectively. Finally, the NAtS-L layers have 12 and 18 softmax attention heads for the 380M and 800M scales. However, since the number of GDN heads does not increase with the hidden states, we group every 2 and 3 softmax attention heads to match each GDN head.

## B. Comparison with other Sparse Attention and Hybrid Attention Architectures

Many different attention architectures are designed to eliminate the quadratic computational complexity of self-attention operations. This include

1. Sparse attention models that assign fixed budgets within each head, e.g., selecting only top-k or the most recent tokens to compute the attention outputs. This model type includes Native Sparse Attention (Yuan et al., 2025) and MOBA (Lu et al., 2025). However, these approaches apply the same budgets to all layers and heads, which may not align with the required semantic information across layers and scenarios.

2. Models that learn an end-to-end sparse model without any hard constraints (Deng & Lindauer, 2025). However, this model uses a sliding window and local attentions to model local information, which might not be flexible enough to capture how much information to preserve at each time step.

3. Models that interleave full attention operations with a cheaper variation. These models replace a fraction of the full attention layers with a cheaper variant, e.g., sparse attention or sliding-window attention (Yang et al., 2025c; Lenz et al., 2025; Kimi-Team, 2025; Qwen-Team, 2025; Ren et al., 2025; Xiaomi, 2026). However, these models still contain full-attention layers that become a bottleneck as the input context length scales.

4. Models that adaptively determine the sparse attention operations using human heuristics or with a learnable process. The former includes approaches that approximate full attention outputs during prefilling (Jiang et al., 2024; Lai et al., 2025) or the token eviction process during decoding stages (Zhang et al., 2023; Xiao et al., 2024; Li et al., 2024), while the latter trains a predictor to determine which attention map to compute (Gao et al., 2024; Xiao et al., 2025; Yang et al., 2025d). However, these approaches aim to recover full-attention outputs; therefore, their performance is bounded by the raw attention models they approximate.

5. Models that apply hybrid softmax-attention and linear attention within the same layer (McDermott et al., 2025; Zhang et al., 2025; Li et al., 2026a). However, these approaches either manually set transition points to switch between softmax and linear attention or apply only linear attention to approximate the full attention output. These heuristics might not be flexible enough to handle input contexts with varying levels of semantic information.

In contrast, NAtS-L adaptively selects optimal tokens for each layer, thereby achieving a better accuracy-efficiency trade-off. Additionally, the token-wise hybrid attention operations force the model to assign long-context-related information to the softmax attention and the remaining parts to the linear attention. This allows softmax attention to focus more on query-related information, thereby providing a potentially stronger model than full attention.

*Table 5.* Evaluation results on retrieval tasks with different sparse attention variations for smaller models

|              | SWDE  | SQD   | FDA   | TQA   | NQ    | DROP  | Avg.  |
|--------------|-------|-------|-------|-------|-------|-------|-------|
| Transformer  | 39.06 | 2.01  | 14.79 | 45.44 | 17.68 | 18.97 | 22.99 |
| NSA          | 31.77 | 10.49 | 6.90  | **46.80** | 17.29 | 19.07 | 22.05 |
| MOBA         | 36.18 | 3.59  | 12.52 | 44.96 | 17.71 | 16.91 | 21.98 |
| GDN SWA      | 8.28  | 8.31  | 0.54  | 24.35 | 5.04  | 4.65  | 8.53  |
| FlexPrefill  | 37.17 | 1.94  | 10.25 | 45.50 | 9.69  | **19.69** | 20.71 |
| SnapKV       | 33.12 | 2.35  | 10.44 | 42.36 | 8.93  | 14.66 | 18.64 |
| AdaKV        | 31.23 | 2.98  | 9.35  | 41.00 | 8.93  | 14.28 | 17.96 |
| PyramidKV    | 33.93 | 2.58  | 10.62 | 39.99 | 9.09  | 14.23 | 18.41 |
| NAtS-L       | **44.82** | **32.98** | 13.07 | 45.02 | 18.02 | 16.96 | 28.48 |
| NAtS-L Hybrid | 44.37 | 32.91 | **42.20** | 46.15 | **20.72** | 18.78 | **34.19** |

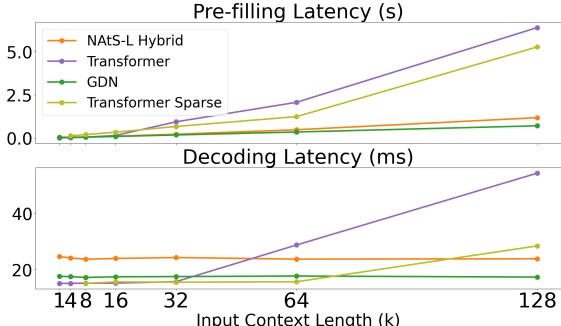

*Figure 7.* Inference Latency with training-free sparse attention during pre-filling with FlexPrefill (Lai et al., 2025) and decoding stage with SnapKV (Li et al., 2024)

We further compare NAtS-L with other variations on the retrieval tasks: NSA (Yuan et al., 2025), MOBA (Lu et al., 2025), and Gated Delta Net with sliding-window attention with size 1024 across all layers (GDN SWA). Additionally, we also compare them against the other training-free approaches during training or decoding: FlexPrefill (Lai et al., 2025), SnapKV (Li et al., 2024), AdaKV (Feng et al., 2025), PyramidKV (Cai et al., 2025). We evaluate all approaches in the setting with a 380M model size and require sparse attention variations to use only 50% of the computational budget. The result is shown in Table 5. Overall, all sparse attention variations achieve similar performance to full attention models, while both NAtS-L and NAtS-L Hybrid provide a stronger retrieval ability, showing that a token-wise mixture of linear and softmax attention provides a model with higher retrieval ability. We also compare the inference latency between NAtS-L Hybrid and training-free transformer acceleration approaches during pre-filling (FlexPrefill (Lai et al., 2025)) and decoding stages (SnapKV (Li et al., 2024)). Overall, NAtS-L Hybrid achieves a lower computational cost during the pre-filling stage, even with a sparse transformer variant and faster decoding, compared to the KV evication baselines at the 128$k$ context length, further demonstrating the efficiency of the NAtS-L approach.

## C. Additionally Experiment Results

### C.1. Results on small-scale models

In Section 5.1, we presented the main results with 800M model scales trained in 50B tokens. In this section, we present additional results for the 380M model scale trained on 15B tokens.

Table 6 shows the performance on the smaller scale models. All the models still perform close on the zero-shot benchmark, while NAtS-L Hybrid and NAtS-L achieve the optimal performance on the retrieval benchmarks.

Figure 8 illustrates the extrapolation ability of different smaller models; the overall results still confirm the conclusion in Figure 3: NAtS-L Hybrid and NAtS-L could efficiently extrapolate towards the unseen context length.

Table 7 and 8 show the result on Longbench and RULER benchmark; the result is still consistent with the larger model

*Table 6.* Evaluation results on language modeling and retrieval tasks for smaller models.

| Model | Wiki. ppl ↓ | LMB. ppl ↓ | LMB. acc ↑ | PIQA acc ↑ | Hella. acc ↑ | Wino. acc ↑ | OQA. acc ↑ | ARC-c acc ↑ | ARC-e acc ↑ | Avg. | SWDE | SQD | FDA | TQA | NQ | DROP | Avg. |
|---|---|---|---|---|---|---|---|---|---|---|---|---|---|---|---|---|---|
| GDN | 28.76 | 49.19 | 27.13 | 65.51 | 37.88 | 50.36 | 31.40 | 27.39 | 54.84 | 42.07 | 16.92 | 28.08 | 8.17 | 45.14 | 9.31 | 17.06 | 20.78 |
| Mamba2 | 29.36 | 53.53 | 27.05 | _66.65_ | **38.39** | **52.41** | _32.00_ | 26.19 | **56.69** | 42.77 | 13.77 | 27.92 | 4.26 | 44.91 | 9.06 | 15.57 | 19.25 |
| Transformer | 29.42 | 44.12 | **31.03** | 65.78 | 37.14 | 51.14 | **33.60** | **28.24** | 55.89 | **43.26** | 39.06 | 2.01 | _14.79_ | 45.44 | 10.01 | **18.97** | 21.71 |
| GDN Hybrid | _28.37_ | **40.08** | 30.64 | 65.34 | 37.67 | _52.01_ | 31.80 | 27.47 | 56.36 | 43.04 | 33.30 | **34.42** | 9.35 | _45.73_ | 9.98 | 17.06 | 24.97 |
| NAtS-L | 28.52 | 52.32 | 27.56 | 65.34 | 37.53 | 51.93 | 31.60 | 27.13 | 55.39 | 42.36 | **44.82** | _32.98_ | 13.07 | 45.02 | _10.14_ | 16.96 | _27.16_ |
| NAtS-L Hybrid | **27.88** | _40.63_ | _30.97_ | **66.65** | _38.04_ | 50.28 | 31.20 | _27.56_ | _56.61_ | _43.04_ | _44.37_ | 32.91 | **42.20** | **46.15** | **16.53** | _18.78_ | **33.49** |

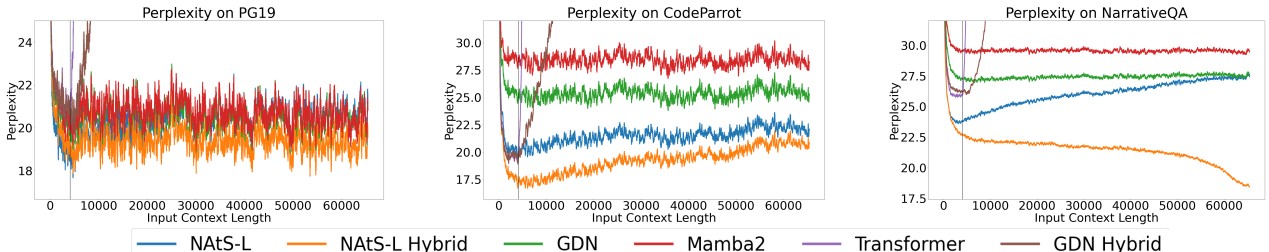

*Figure 8.* Per-token perplexity on different datasets on a smaller model size, all the models are trained with 4096 tokens (the black vertical line)

cases.

## C.2. Mixture of SoftmaxAttention and Mamba2 architecture

NAtS-L does not specify the operations applied to linear attention. Here, we propose a variation of NAtS-L that searches for the softmax attention-Mamba2 architecture (Dao & Gu, 2024). The result is shown in Table 9. By intersecting softmax-attention into Mamba2 operations, NAtS-L can also enhance the retrieval ability of the Mamba2 model, showing that NAtS-L can be generalized to different linear attention and softmax attention variations.

## C.3. Task-Wise Results on RULER

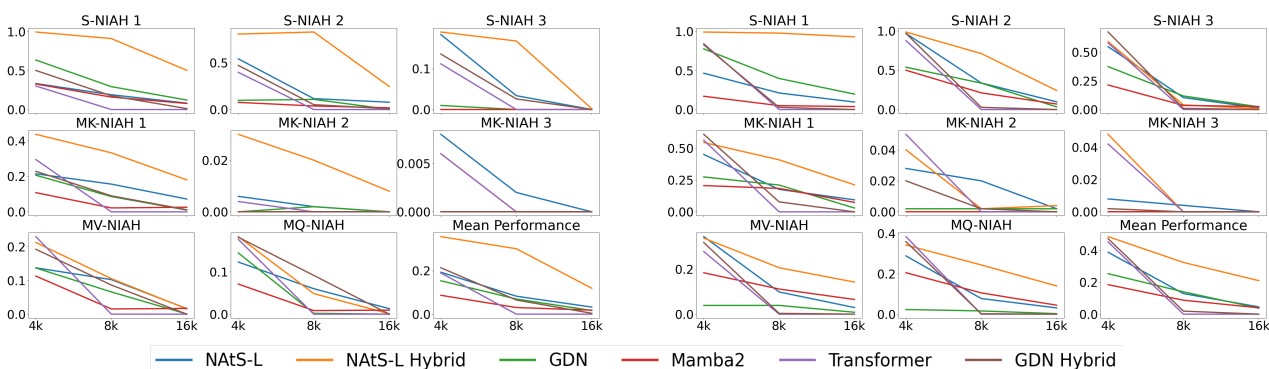

*Figure 9.* Per-Task Ruler performance. (Left) RULER performance with $380M$ parameters. (Right) RULER results with $800M$ parameters

We show the overall mean scores of the RULER benchmark in Table 2. The per-task performance is illustrated in Figure 9. Overall, NAtS-L performs better on the single needle-in-a-haystack tasks and achieves a performance comparable to other baselines on the other tasks. Additionally, while the other baselines might fail when the input context length exceeds $4k$, NAtS-L Hybrid can still maintain its performance across different scales.

*Table 7.* Evaluation results for LongBench benchmarks on smaller models with input context length below $4k$.

|  | 2WM | HQA | MFQ | QQA | MN | GOV | TRC | TQA | SSM | LCC | RBP |
|---|---|---|---|---|---|---|---|---|---|---|---|
| GDN | 7.02 | 6.03 | 12.58 | 2.57 | 9.25 | 8.09 | 21.50 | 13.18 | **10.47** | 12.87 | 14.98 |
| Mamba2 | 6.25 | 4.88 | 11.72 | 2.40 | 6.48 | 7.15 | 12.00 | 11.83 | 3.17 | 18.32 | 17.62 |
| Transformer | 3.73 | 3.14 | 9.79 | 1.97 | 8.57 | 8.14 | 13.00 | 10.18 | 5.49 | 17.57 | 10.42 |
| GDN Hybrid | **10.46** | 4.56 | 11.72 | 2.28 | **11.10** | 8.99 | **31.00** | **13.52** | 5.37 | 17.89 | 16.46 |
| NAtS-L | 9.86 | **6.20** | 13.11 | 2.77 | 10.36 | **8.99** | 18.00 | 11.81 | 8.00 | 13.65 | 17.50 |
| NAtS-L Hybrid | 6.24 | 6.10 | **13.34** | **3.82** | 10.14 | 6.81 | 26.00 | 12.89 | 8.67 | **19.62** | **20.50** |

*Table 8.* Mean Scores for RULER benchmarks with different input context length with smaller model size.

|  | 4k | 8k | 16k |
|---|---|---|---|
| GDN | 0.15 | 0.07 | 0.02 |
| Mamba2 | 0.09 | 0.03 | 0.02 |
| Transformer | 0.19 | 0.00 | 0.00 |
| GDN Hybrid | 0.21 | 0.07 | 0.00 |
| NAtS-L | 0.19 | 0.08 | 0.03 |
| NAtS-L Hybrid | **0.36** | **0.30** | **0.12** |

## C.4. Additional Results on Longbench

Table 10 and 11 illustrate the results on the longbench when the input context goes beyond $4k$. NAtS-L and NAtS-L Hybrid can efficiently extrapolate their performance towards the input context tokens beyond their training context length.

## C.5. Token Types Distribution for NAtS-L

We show the distributions of the token types for NAtS-L Hybrid in Figure 5. Here, we illustrate the task-wise experimental results. As shown in Figure 10, the softmax attention token distributions generally follow a similar trend, with some of the heads might change their roles given the input context. However, the shallower layers might contain more linear attention heads while the softmax attention heads might be located more in the in the intermedaite and deeper layers. This indicates that the model tends to construct local correlation in the earlier layer and then gradually switch to global correlation in the deeper layer. This might also provide further insights into the design of the new hybrid attention architectures. i.e., rather than assigning softmax attention layers uniformly to the model, we should place the softmax layers more towards the deeper layers.

Figure 6 illustrates the Token distributions for each time step on the PG 19 dataset. Figure 11 shows the token distribution of the other two datasets. Overall, the trends are similar to those on the PG19 dataset for most heads when they are dominated by a single operation. However, for heads that contain both softmax attention and linear attention layers, the distribution can be quite different. For instance, the model evaluated on NarrativeQA has more GDN operations on head 2 of layer 16, while more softmax attention is assigned to this head when evaluated on the CodeParrot task, indicating that NAtS-L can adapt its budgets to different tasks based on the input context.

Figure 12 illustrates the output weight distributions across different softmax attention heads, where the weights are computed by Equation 22. We can see that different operations contribute differently across heads. Their weights change over time, indicating the need to introduce time-dependent attention weights to balance the contributions of different attention operations.

*Table 9.* Comparison between mixing softmax attention operations and Mamba2 operations with NAtS-L and the Mamba2 model.

|  | SWDE | SQD | FDA | TQA | NQ | DROP | Avg. |
|---|---|---|---|---|---|---|---|
| NAtS-L Mamba2 | **32.58** | 21.18 | **32.30** | 41.35 | **12.48** | 12.12 | **25.34** |
| Mamba2 | 13.77 | **27.92** | 4.26 | **44.91** | 9.06 | **15.57** | 19.25 |

*Table 10.* Evaluation results for LongBench benchmarks with input context length below $8k$.

| Model | 2WM | HQA | MFQ | QQA | MN | GOV | TRC | TQA | SSM | LCC | RBP |
|---|---|---|---|---|---|---|---|---|---|---|---|
| *800M Parameters* | | | | | | | | | | | |
| GDN | 8.01 | 5.41 | 13.11 | 5.06 | **10.13** | 6.82 | 28.00 | 22.43 | **19.86** | 18.35 | 16.23 |
| Mamba2 | 8.84 | 4.92 | 13.47 | 4.64 | 8.25 | 7.40 | 4.00 | 19.52 | 10.81 | **21.68** | **21.94** |
| Transformer | 0.52 | 0.32 | 2.06 | 2.03 | 4.54 | 4.39 | 0.00 | 0.31 | 3.41 | 4.80 | 4.56 |
| GDN Hybrid | 5.00 | 2.93 | 6.56 | 3.35 | 4.17 | 5.41 | 19.00 | 12.83 | 1.76 | 13.27 | 10.28 |
| NAtS-L | 7.02 | 4.87 | 12.50 | **5.19** | 7.77 | 8.07 | **33.00** | 22.35 | 19.21 | 16.33 | 16.35 |
| NAtS-L Hybrid | **11.47** | **5.52** | **13.92** | 2.96 | 9.99 | **13.46** | 17.00 | **32.17** | 19.05 | 10.28 | 13.82 |
| *380M Parameters* | | | | | | | | | | | |
| GDN | 5.89 | 4.17 | 10.82 | 3.45 | 8.33 | **7.31** | 26.00 | **13.86** | 9.05 | 12.93 | 12.45 |
| Mamba2 | 5.30 | 3.27 | 9.86 | 2.92 | 7.65 | 5.72 | 9.00 | 12.07 | 3.94 | 17.28 | 13.04 |
| Transformer | 1.30 | 1.38 | 2.83 | 2.57 | 5.38 | 4.19 | 0.00 | 2.80 | 1.72 | 7.65 | 7.65 |
| GDN Hybrid | 6.00 | 2.61 | 6.39 | 2.93 | 8.17 | 6.59 | 17.00 | 10.97 | 4.76 | 15.35 | 14.56 |
| NAtS-L | **7.69** | **4.92** | 9.75 | 3.57 | **8.85** | 6.29 | 13.00 | 13.52 | 6.34 | 11.92 | 14.72 |
| NAtS-L Hybrid | 5.93 | 3.85 | **11.63** | **4.24** | 6.86 | 4.72 | **39.00** | 11.52 | **9.53** | **17.85** | **18.40** |

*Table 11.* Evaluation results for LongBench benchmarks with input context length beyond $8k$.

| | 2WM | HQA | MFQ | QQA | MN | GOV | TRC | TQA | SSM | LCC | RBP |
|---|---|---|---|---|---|---|---|---|---|---|---|
| *800M Parameters* | | | | | | | | | | | |
| GDN | 6.18 | **6.01** | 11.59 | 2.76 | **9.18** | 5.21 | **24.00** | 19.86 | **21.41** | 16.32 | 17.50 |
| Mamba2 | **6.90** | 5.09 | 11.14 | 2.60 | 7.71 | 6.52 | 2.00 | 19.12 | 13.36 | **18.85** | **21.24** |
| Transformer | 0.55 | 0.39 | 1.18 | 1.73 | 1.68 | 2.73 | 0.00 | 1.40 | 0.00 | 4.08 | 3.57 |
| GDN Hybrid | 1.84 | 1.77 | 4.13 | 1.13 | 2.08 | 6.63 | 10.00 | 6.83 | 0.32 | 10.80 | 9.32 |
| NAtS-L | 4.24 | 5.21 | 13.09 | **3.31** | 8.04 | 7.92 | 21.00 | 23.29 | 19.77 | 16.31 | 17.24 |
| NAtS-L Hybrid | 6.66 | 5.72 | **13.36** | 1.82 | 9.01 | **9.10** | 17.00 | **28.60** | 19.85 | 13.31 | 11.54 |
| *380M Parameters* | | | | | | | | | | | |
| GDN | 4.34 | 4.28 | 10.81 | 2.59 | 7.92 | 6.14 | 19.00 | 15.09 | 8.09 | 12.53 | 13.85 |
| Mamba2 | 4.00 | 3.60 | 10.62 | 1.17 | 7.88 | 5.96 | 13.00 | 12.25 | 2.26 | **17.24** | 14.75 |
| Transformer | 0.58 | 0.61 | 0.62 | 1.77 | 4.15 | 4.23 | 1.00 | 2.74 | 4.71 | 8.53 | 6.50 |
| GDN Hybrid | 1.93 | 1.69 | 6.95 | 1.40 | 7.68 | **7.29** | 7.00 | 9.53 | 2.45 | 15.01 | 15.06 |
| NAtS-L | **4.97** | **4.89** | 9.95 | 1.81 | **8.61** | 6.29 | 12.00 | **15.12** | 5.50 | 14.55 | 12.09 |
| NAtS-L Hybrid | 3.08 | 4.56 | **11.49** | **2.79** | 6.80 | 3.87 | **33.00** | 11.70 | **10.43** | 16.22 | **17.60** |

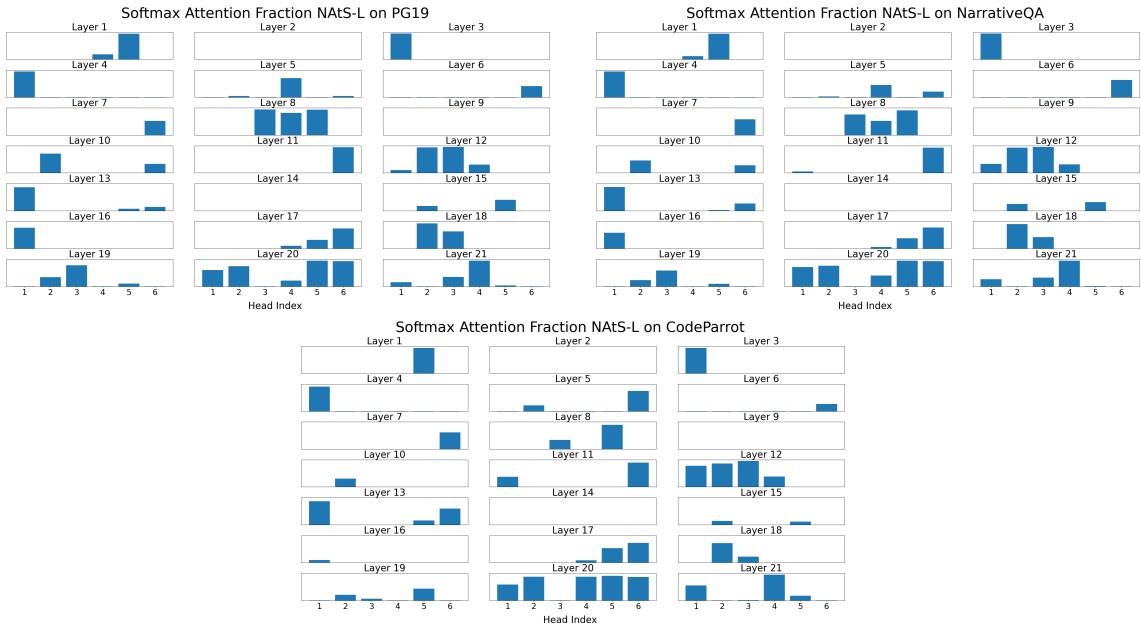

*Figure 10.* Token number distributions of NAtS-L on PG19, NarrativeQA, and CodeParrot dataset

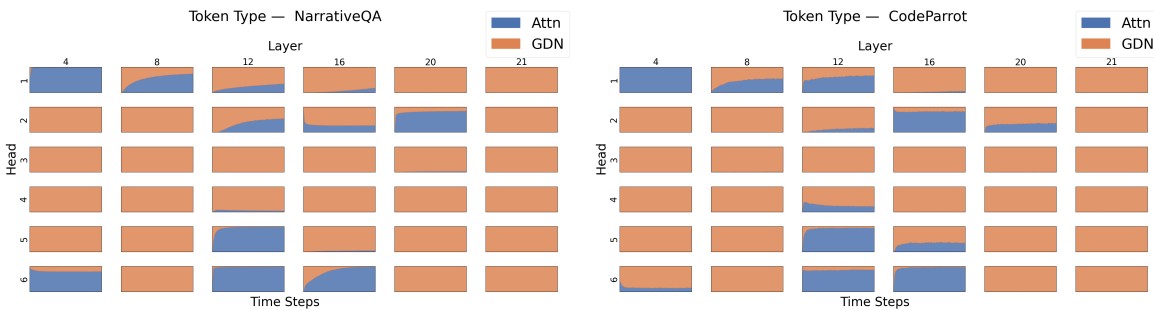

*Figure 11.* Time-Wise Token distributions for NarrativeQA and CodeParrot dataset.

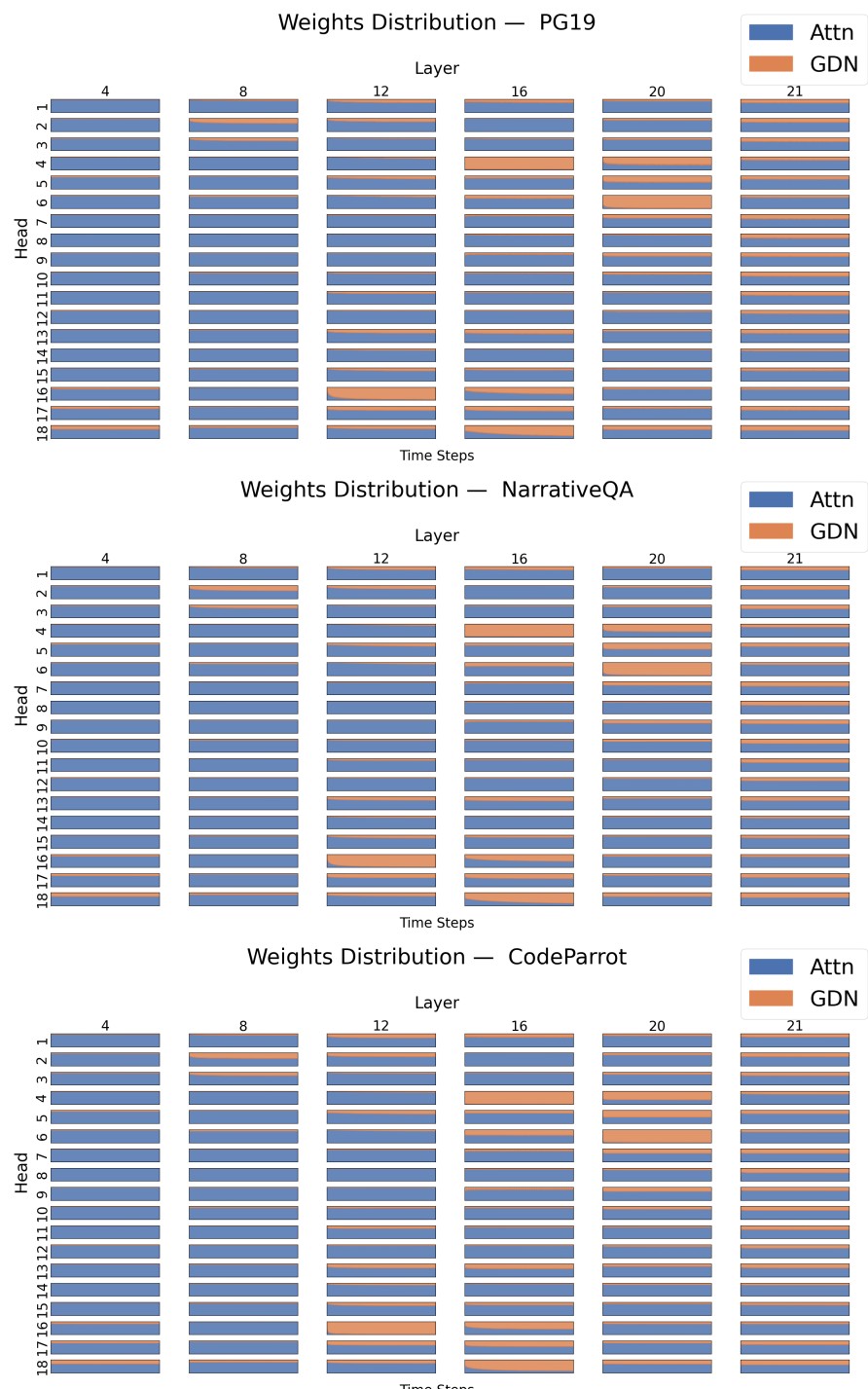

*Figure 12.* Time-Wise Attention Output Weight Distributions on PG19, NarrativeQA, and CodeParrot dataset.

