# OpenReview forum: "Neural Attention Search Linear: Towards Adaptive Token-Level Hybrid Attention Models"
_ICML.cc/2026/Conference — ICML 2026 regular_

### Official Review · Reviewer_BR91 · 2026-02-24

**Soundness:** 3
**Presentation:** 3
**Significance:** 3
**Originality:** 3
**Overall Recommendation:** 4
**Confidence:** 4

**Summary:**

The paper proposes NAtS-L, a chunk-level hybrid attention method that dynamically routes chunks of tokens to either softmax or linear (Gated DeltaNet) attention within the same layer, using an input-dependent learned scoring mechanism. The central claim is that this chunk-level flexibility offers better efficiency-performance trade-offs than layer-level hybrid architectures.

**Compliance With Llm Reviewing Policy:**

Affirmed.

**Final Justification:**

The paper presents a sound and well-motivated chunk-level hybrid attention mechanism with solid empirical results. The rebuttal fully resolved my concerns. My weak accept reflects the somewhat incremental nature of the contribution. This is a sensible and useful extension of existing hybrid attention architectures rather than a fundamental advance, but it makes a real contribution to efficient long-context modelling and provides a foundation for future work.

**Key Questions For Authors:**

1. **Regarding learned weight distributions:** Could you provide the magnitude distributions or summary statistics for the learned weights in Eq. 20? Specifically, does the model utilise both components across all layers, or does it converge toward one mechanism?
2. **Regarding chunk size:** What were the exact chunk sizes used for each experiment? Additionally, how does varying the chunk size impact the routing behaviour and the overall compute required?
3. **Regarding aggregate softmax usage:** What is the model-level aggregate percentage of chunks routed to softmax attention across the different benchmarks and context lengths? Is there a risk of the model defaulting to full softmax attention in certain regimes?
4. **Regarding Eq. 8 dimensionality:** Could you clarify the exact shapes of $W^{\text{score}}$ and $\text{score}_t$?

**Limitations:**

yes

**Strengths And Weaknesses:**

**Strengths**

The main strength of the paper lies in its introduction of an adaptive chunk-level hybrid attention mechanism that combines softmax and linear attention. The learnable neural attention search strategy removes reliance on fixed architectural heuristics and allows the model to allocate computationally expensive softmax attention only when needed, improving scalability for long contexts. The framework is implemented with practical design choices such as shared QKV projections and chunk-wise computation, keeping parameter overhead low while remaining hardware-friendly. Extensive experiments across language modelling, retrieval, and long-context benchmarks demonstrate consistent robustness, improved extrapolation behaviour, and lowered latency compared to other methods.

**Weaknesses**

1. **Lack of empirical verification for hybrid balance:** Prior work [1, 2] suggests that hybrid architectures often over-rely on the softmax component unless explicitly constrained. Because Eq. 20 uses unconstrained weights, it is unclear if the model is truly utilising both components meaningfully. The absence of weight distribution statistics makes it difficult to verify if the hybrid nature is being maintained or if the model is collapsing into a single mode.
2. **Missing details on chunking:** The choice of chunk size is a key hyperparameter that influences routing granularity, hardware efficiency, and long-context capturing. However, the paper does not explicitly report the exact chunk sizes used for the main experiments, nor does it provide an ablation or justification for these choices.
3. **Ambiguity in efficiency guarantees:** The paper’s efficiency claims rely on the assumption that a significant portion of computation is routed away from softmax. However, without an auxiliary loss or aggregate model-level statistics on softmax routing, there is no guarantee that the model won't drift toward heavy softmax usage (and thus quadratic cost) in certain regimes. Currently, the analysis is restricted to localised layers and heads (Figures 5 and 8) rather than a holistic efficiency summary.
4. **Lack of clarity in Eq. 8:** The shape of $W^{\text{score}}$ and the dimensionality of $\text{score}_t$  in Eq. 8 are never stated, making it unclear whether scores are produced per chunk, per head, or per operation type. This ambiguity makes the method difficult to reproduce and understand, especially given that this scoring mechanism is central to the paper's contribution.

Minor formatting note: ICML style guidelines typically place table captions above tables. The authors may want to adjust this in the final version.


[1] Martin Benfeghoul, Teresa Delgado, Adnan Oomerjee, Haitham Bou Ammar, Jun Wang, Zafeirios Fountas. Untangling Component Imbalance in Hybrid Linear Attention Conversion Methods.

[2] Cabannes Loïc, Maximilian Beck, Maria Lomeli, Gergely Szilvasy, Matthĳs Douze, Jade Copet, Pierre-Emmanuel Mazare, Gabriel Synnaeve, Herve JegouShort Window Attention Enables Long-Term Memorization. In The Fourteenth International Conference on Learning Representations 2026 .

---

> ### Author Rebuttal · Authors · 2026-03-31
>
> Dear Reviewer BR91,
>
> we thank you for your constructive feedback.  We also thank you for pointing out the strength of NAtS-L architecture.
>
> Regarding your concerns:
>
> > Lack of empirical verification for hybrid balance
>
>  Indeed, the weights are normalized with token-level softmax (meaning that $w^{nla}t+w^{la}t=1$). We will make this decision clearer, we also make an assessment on the impact of this decision, we compare with another two variatiosn: weights without activation function and sigmoid function:
>
> |                |   SWDE |   SQD |   FDA |   TQA |    NQ |   DROP |   Avg. |
> |:---------------|-------:|------:|------:|------:|------:|-------:|-------:|
> | NAtS-L Softmax |  44.37 | 32.91 | 42.2  | 46.15 | 16.53 |  18.78 |  33.49 |
> | NAtS-L Sigmoid |  44.91 | 28.65 | 31.58 | 44.37 | 14.76 |  15.81 |  30.02 |
> | NAtS-L No Act  |  41.76 | 33.04 | 28.86 | 47.75 | 14.41 |  17.59 |  30.57 |
>
> NAtS-L with softmax (our current version) outperforms the other two variations.
>
> > The absence of weight distribution statistics
>
> Thanks for pointing this out. We study the distribution of the weights in https://drive.google.com/file/d/1jNmk-qFykNFj818FisCace5g2GhA3pdA/view?usp=sharing , Figures 2 and 3. We could see that not all the heads rely only on softmax attention layers. Many heads might still emphasize linear attention operations. Additionally, shallower layers tend to focus more on the softmax attention operations, while the attention patterns in the deeper layers are more mixed.
>
> > Missing details on chunking
>
> We thank the reviewer for pointing out this uncertainty. We briefly discussed the chunk size in lines 243-246. For efficiency, we set the NAtS chunk size to be no less than the GDN chunk size.  In the meantime, we want the chunk size to be as small as possible to be more flexible to different input contexts. Following the GDN implementation, we use a chunk size of 64 for NAtS-L to have a better trade-off between model performance and training efficiency. We also made an ablation study with an increased chunk size:
>
> |                  |   SWDE |   SQD |   FDA |   TQA |    NQ |   DROP |   Avg. |
> |:-----------------|-------:|------:|------:|------:|------:|-------:|-------:|
> | NAtS-L Chunk 64  |  44.37 | 32.91 | 42.2  | 46.15 | 16.53 |  18.78 |  33.49 |
> | NAtS-L Chunk 128 |  30.78 | 28.42 | 32.85 | 45.32 | 10.86 |  18.88 |  27.85 |
>
> > Ambiguity in efficiency guarantees
>
> In this paper, we do not regularize the fraction of softmax attention and linear attention to explore how the model would react to the input context with only gradient information, without any human intervention. In our evaluation, given the current next-token prediction scenario, we found that the model tends to use more linear attention than softmax attention. As discussed in Figures 5 and 8 of the paper, some heads contain only GDN operations, while no head contains a pure-softmax attention operation.
>
> Additionally, if the optimizer pushes all tokens to the softmax attention operation, this indicates that the pure softmax attention model remains the strongest long-context modeling model; it would then be intended that NAtS-L represents this, as the core contribution is selecting the attention type with respect to the input context and pretraining scenarios. Hence, in this paper, we will first let the model explore the potential optimal attention operations without any auxiliary losses.
>
> > Regarding aggregate softmax usage
>
> The result is illustrated in Figures 4 and 5 under https://drive.google.com/file/d/1jNmk-qFykNFj818FisCace5g2GhA3pdA/view?usp=sharing . Interestingly, the model tends to select more GDN blocks. In the earlier stage, only a few softmax attention heads are selected. Additionally, we could see that sometimes the fraction of softmax attention increases or decreases as the time steps grows. While on certain heads, the fraction of the two operations might be different (head 5 of layer 0, head 1 of layer 4). This shows that NAtS-L can adaptively adjust the token types given different input contexts.
>
> > lack of clarity in Eq. 8:
>
> We thank the reviewer for pointing out this lack of clarity. The weights $w_{score}$ are applied to each pooled token, and therefore have a shape of  $w_{score} \in R^{d \times n_{heads} \times n_{opts}}$ where $d$ is the hidden dimension of the input feature map. The score is applied to each chunk for each head individually, i.e., $score_t \in R^{1 \times n_{heads} \times n_{opts}}$ for each chunk. So for a sequence of length L, we have $score \in R^{L/C \times n_{heads} \times n_{opts} }$ where C is the chunk size. ,
>
> > Minor formatting note: ICML style guidelines typically place table captions above tables.
>
> Thank you for pointing this out. We will update our manuscript accordingly.

---

> > ### Author Rebuttal · Reviewer_BR91 · 2026-04-01
> >
> > Thank you for the detailed rebuttal. My concerns have been addressed, and I maintain my current score.

---

> > > ### Author Response · Authors · 2026-04-01
> > >
> > > Dear Reviewer BR91,
> > >
> > > We are glad that we have addressed your concerns, and thank you for the positive score!

---

### Official Review · Reviewer_W8we · 2026-02-27

**Soundness:** 3
**Presentation:** 3
**Significance:** 3
**Originality:** 3
**Overall Recommendation:** 4
**Confidence:** 3

**Summary:**

This work proposes to combine the attention mechanism and linear attention. The way is compute for attention and linear attention separately, and then combine with the weight from the projection of the query embedding.

**Compliance With Llm Reviewing Policy:**

Affirmed.

**Final Justification:**

The rebuttal address the concerns so that I keep the score.

**Key Questions For Authors:**

N/A

**Strengths And Weaknesses:**

Strength:
* The proposed idea is clear and easy to understand.
* The paper is well-written with Figure 2.
* The experiment results support the core claim of this work.

Weakness:
* In Table 1, the GDN hybrid is better than the proposed method. For example, GDB Hybrid achieves 68.18, while NAtS-L Hybrid achieves 66.75.
* For Equation 20, what is the shape of $w_t^{nla}$ and $w_t^{la}$?
* How about modeling the two operations with MoE? For example, mapping the query embedding to 2-dimension, and then using softmax router, sigmoid router, or KERN router to combine the attention and GDN?
* For Table 2, the proposed method has better performance than the Transformer. Could you explain the hyperparameter setting? Do they has the same hidden size, attention head number of the attention operation?

This work could be regarded as a special type of MoE, where one expert is GDN, and another expert is Attention. The major novelty is: the attention and GDB share the same query, key and value, and it works.

---

> ### Author Rebuttal · Authors · 2026-03-31
>
> Dear Reviewer W8we,
>
> we thank you for your constructive feedback.  We also thank you for pointing out that the paper is well written and easy to follow.
> Regarding your concerns:
>
> > In Table 1, the GDN hybrid is better than the proposed method.
>
> The results in Table 1 primarily focus on zero-shot common-sense tasks and place less emphasis on long-context modeling. Since different architectural designs mainly influence the ability to model in-context correlation, it is possible that NAtS-L could be outperformed by some other models on certain tasks. However, the overall result still shows that NAtS-L achieves the best average performance (including the two perplexity evaluation tasks). This still shows the strength of NAtS-L on the common sense benchmarks.
>
> > For Equation 20, what is the shape of $w_t^{nla}$ and $w_t^{la}$?
>
> $w_t^{nla}$ and $w_t^{la}$ are time token-wise weights that are assigned to each head. So, assuming that we have an input sequence of length $L$ and the architecture has $H $attention heads (as discussed in lines 251-256, the number of softmax attention heads is usually larger than the GDN head number), both values will be of size $R^{L \times H}$. This value is still minimal compared to the QKV values in transformer models. We will update our manuscript to make this clearer.
>
> > How about modeling the two operations with MoE?
>
> Thanks for the advice; indeed, the weights in Equation 20 are normalized using a softmax function. We also evaluated the models whose weights are not normalized or weighted with sigmoid:
>
> |                |   SWDE |   SQD |   FDA |   TQA |    NQ |   DROP |   Avg. |
> |:---------------|-------:|------:|------:|------:|------:|-------:|-------:|
> | NAtS-L Softmax |  44.37 | 32.91 | 42.2  | 46.15 | 16.53 |  18.78 |  33.49 |
> | NAtS-L Sigmoid |  44.91 | 28.65 | 31.58 | 44.37 | 14.76 |  15.81 |  30.02 |
> | NAtS-L No Act  |  41.76 | 33.04 | 28.86 | 47.75 | 14.41 |  17.59 |  30.57 |
>
> NAtS-L with softmax (our current version) outperforms the other two variations.  We will make this decision process clearer in our manuscript.
>
> > Could you explain the hyperparameter setting?
>
> We list the architecture hyperparameter settings in Appendix A.1. Basically, the NAtS-L models share the same architecture as GDN to ensure that the GDN parts of the architecture also perform well, though they do not have the same number of heads or hidden dimensions as the transformer model. However, the GDN has the same architecture as NAtS-L backbone, but it still cannot perform as well as the transformer architecture in long-context modeling tasks. We could state that the benefit does not come from the architecture backbone, but the hybrid attention operator that constructs corrections between different tokens.

---

> > ### Author Rebuttal · Reviewer_W8we · 2026-04-01
> >
> > Thank you for the response. There are no further concerns, so that I keep the score.

---

> > > ### Author Response · Authors · 2026-04-01
> > >
> > > Dear Reviewer W8we,
> > >
> > > We are glad that we have addressed your concerns, and thank you for the positive score!

---

### Official Review · Reviewer_YKJN · 2026-03-04

**Soundness:** 3
**Presentation:** 2
**Significance:** 3
**Originality:** 2
**Overall Recommendation:** 4
**Confidence:** 3

**Summary:**

The authors introduce Neural Attention Search Linear (NAtS-L), a framework that learns to apply linear and softmax attention to different parts of a sequence.

**Compliance With Llm Reviewing Policy:**

Affirmed.

**Final Justification:**

Most major concerns were addressed, thus I have slightly increased my score to reflect this.

**Key Questions For Authors:**

There's some rather important details that are missing that hinder the clarity of the work.

- At inference time, when tokens are generated auto-regressively, how does this affect the way in which the right attention type is selected. From what I understand, the method benefits from the chunk-based approach but this requires multiple tokens and is therefore only applicable during pre-filling.
- While perhaps different from the motivation of the work, the authors don't compare against KV-caching techniques for either pre-filling or decoding. For example, it is well known that numerous method for cache management or pre-filling improve the ability to handle long contexts, many of which do not require any additional training. While these are applicable to transformer models, this might not be the case for NAtS-L, which is a limitation. Furthermore, comparison is against a standard soft-max transformer, but given the wide array of methods for improving latency both during pre-filling and inference it would behoove the authors to provide results that demonstrate the actual ways in which they may be used for long-context inference, as otherwise the comparison remains somewhat moot and unrepresentative of real-world use, which is part of this work's motivation.

**Limitations:**

See above.

**Strengths And Weaknesses:**

In terms of strengths, the experiments do seem to show that the method is effective and the experimental setup appear appropriate for the goal of the work. On one note however, is that results differ from results in the original GDN paper on some fronts, so I do wonder if this is due to re-using a set of hyper-parameters that is functional for NAtS-L but is not ideal for other models.

In terms of weaknesses, there method itself feels more of a mixture of existing methods, as it is simply placing two existing models in parallel within a layer. Such a structure has already been explored in the past in works such as Hymba, so the primary contribution appears to be introducing a chunk-based routing mechanism, which isn't explored in significant detail. Given this, the authors should provide more than just one variant with GDN as the linear-attention module, as this obfuscates whether or not improvements are simply due to this module and not because of the structure itself.

---

> ### Author Rebuttal · Authors · 2026-03-31
>
> Dear Reviewer YKJN,
>
> we thank you for your constructive feedback. We also appreciate that you point out the efficiency of NAtSL.
>
> Regarding your concerns
>
> > results differ from results in the original GDN paper
>
> The results reported in the GDN paper are evaluated using 1.3 B-scale models trained on 100B tokens, whereas we train all models only up to 800M scale with 50B tokens.
>
> >  the method itself feels more of a mixture of existing methods, such a structure has already been explored in the past in works such as Hymba
>
> Previous works, such as Hymba, mostly focus on the head-level hybrid attention, while NAtS-L could adaptively determine the optimal operations for each token chunk. This results in a much more flexible but much harder problem. For instance, for a sequence length of $4096$ and chunk size of $64$, even if each chunk has 2 options, the number of possible operation combinations is still $2^{(4096/64)}=1.8*10^{19}$.  A transformer contains multiple layers with multiple heads in each layer. This could even grow the search space exponentially. Therefore,  searching for the optimal operation types for each chunk is not as easy as selecting the optimal operation for an entire sequence. To the best of our knowledge, we are the first to propose a token-level end-to-end adaptive hybrid model that can be learned from scratch. Other approaches either interleave only between LA and NLA layers, require a pre-trained transformer that can be approximated, or follow a fixed rule with hand-crafted hyperparameters. Therefore, we would argue that NAtS-L is not only a mixture of existing works.
>
> > which isn't explored in significant detail.
>
> We thank you for your feedback. We will update our manuscript to make this clearer.
>
> > authors should provide more than just one variant with GDN
>
> We have an additional experiment searching for the mixture of Mamba2 and transformer models. Results show that NAtS-L, when searched with MAMBA2, can also outperform the MAMBA-only model.
>
> |               |   SWDE |   SQD |   FDA |   TQA |    NQ |   DROP |   Avg. |
> |:--------------|-------:|------:|------:|------:|------:|-------:|-------:|
> | NAtS-L Mamba2 |  32.58 | 21.18 | 32.3  | 41.35 | 12.48 |  12.12 |  25.34 |
> | Mamba2        |  13.77 | 27.92 |  4.26 | 44.91 |  9.06 |  15.57 |  19.25 |
>
> > At inference time, when tokens are generated auto-regressively, how does this affect the way in which the right attention type is selected
>
> NAtS-L can be applied to both the pre-filling and decoding stages. When the NAtS-L does not collect enough tokens in a chunk, all the operations (Linear and Non-Linear attentions) are involved in the computation (as stated in section 4.1, right side of line 166-168).  We update the internal state within each chunk and reroll the linear attention states and KV caches for the corresponding operations when they are not selected for the current chunk.  In Table 4 of the paper, we also show that this approach outperforms applying only GDN or softmax attention to incomplete chunks.
>
> > the authors don't compare against KV-caching techniques for either pre-filling or decoding.
>
> We compare with several prefilling and KV caching techniques such as FlexPrefill, snapKV, AdaKV, and Pyramid KV:
>
> |               |   SWDE |   SQD |   FDA |   TQA |    NQ |   DROP |   Avg. |
> |:--------------|-------:|------:|------:|------:|------:|-------:|-------:|
> | NAtS-L        |  44.82 | 32.98 | 13.07 | 45.02 | 10.14 |  16.96 |  27.16 |
> | NAtS-L Hybrid |  44.37 | 32.91 | 42.2  | 46.15 | 16.53 |  18.78 |  33.49 |
> | Transformer   |  39.06 |  2.01 | 14.79 | 45.44 | 10.01 |  18.97 |  21.71 |
> | FlexPrefill   |  37.17 |  1.94 | 10.25 | 45.5  |  9.69 |  19.69 |  20.71 |
> | AdaKV         |  31.23 |  2.98 |  9.35 | 41    |  8.93 |  14.28 |  17.96 |
> | PyramidKV     |  33.93 |  2.58 | 10.62 | 39.99 |  9.09 |  14.23 |  18.41 |
> | SnapKV        |  33.12 |  2.35 | 10.44 | 42.36 |  8.93 |  14.66 |  18.64 |
>
> We also show the prefilling and decoding time with these techniques (FlexPrefill for pre-filling and snapKV for decoding in https://drive.google.com/file/d/1jNmk-qFykNFj818FisCace5g2GhA3pdA/view?usp=sharing , Figure 1). With $50\\%$ of the KV budget, the sprase transformer achieves a similar speed up as NAtS-L
>
> However, the KV-caching and pre-filling techniques are designed to approximate the full transformer output and, therefore, remain bounded by the softmax attention only models' ability. On the other hand, NAtS-L is not only designed for efficiency, but also to demonstrate the strength of token-level hybrid attention models, as evidenced by our experiments: our NAtS-L token-level hybrid attention model achieves higher retrieval scores than the softmax attention and layer-wise hybrid attention models. Additionally, we would argue that NAtS-L is not orthogonal to these approaches; they can still be applied to accelerate the softmax attention branch of NAtS-L to further reduce the computational costs.

---

> > ### Author Rebuttal · Reviewer_YKJN · 2026-04-01
> >
> > I think my questions have been adequately resolved; there's still some outstanding details regarding possible scalability of the architecture to larger sizes and ensuring that implementations of kernels are flexible for these settings as well, however I do agree that such things are not disqualifying details. I have raised my score accordingly.

---

> > > ### Author Response · Authors · 2026-04-01
> > >
> > > Dear Reviewer YKJN,
> > >
> > > We are glad that we have addressed your concerns, and thank you for raising your score!

---

### Official Review · Reviewer_M6Vf · 2026-03-10

**Soundness:** 3
**Presentation:** 2
**Significance:** 3
**Originality:** 2
**Overall Recommendation:** 4
**Confidence:** 4

**Summary:**

This paper combines the advantages of efficient sparse attention and linear attention to achieve better long-sequence performance, particularly demonstrating meaningful improvements in retrieval tasks and extrapolation capability. The proposed method employs a routing approach (NSA or DSA, as formulated in Equation 8) to classify chunked tokens. A portion of tokens follows the linear modeling approach formulated in Equation 9, while the remaining portion follows the retained nonlinear Softmax modeling approach formulated in Equation 10. By integrating information from both components, the method achieves commendable commonsense reasoning, retrieval, and extrapolation capabilities. The efficiency does not lag significantly behind pure linear models.

**Compliance With Llm Reviewing Policy:**

Affirmed.

**Final Justification:**

I will keep the score at 4. Please refer to our discussion for the details.

**Key Questions For Authors:**

1. The paper does not specify the chunk sizes and quantities used for the NLA and LA portions. While it can be inferred that all selected chunks in the LA portion can be merged into a fixed-size state, the number and size of chunks used in the NLA portion remain unclear. Furthermore, the selection process and the merging process for selected chunks are not described in sufficient detail.

2. Based on the above, the computational complexity of the selection or routing mechanism needs to be discussed, ideally as part of the overall complexity analysis, so that the end-to-end computational complexity of the entire process can be provided, rather than reporting only the complexity of the actual NLA and LA attention computations. Additionally, please explain whether Figure 4 aligns with the theoretical complexity analysis.

3. Include fairer experimental comparisons. Since this paper employs a sparse attention branch, which based on my experience plays a very significant role, comparisons should not be limited to less competitive linear methods. Comparisons with methods such as NSA and MoBA would be appropriate. Furthermore, for GDN-based hybrid models, comparisons should be made at the token-level hybrid within the same layer (i.e., sliding window + GDN), similar to approaches like Based, LoLCATs, and MVA, to ensure consistency with the proposed method.

**Overall Assessment:**

In summary, the tasks selected in this paper are well-chosen, and the authors present a substantial amount of meaningful results. If the above questions and concerns can be addressed, I would be pleased to raise my score to 4. Additionally, the following points may require more extensive discussion and could be considered beyond the scope of the current revision:

4. There are many methods that combine sparse attention with linear attention, and a more reasonable explanation and combination might emerge from considering token-level multi-scale responsibilities.

5. Regarding efficient implementation, it would be helpful to discuss how the selection mechanism and attention computation are integrated—whether they are implemented separately using FLA's GDN operator and FlashAttention operator and then combined through simple PyTorch operations.

These additional points do not affect my current score, but I hope they may help make the paper more complete.

**Strengths And Weaknesses:**

**(1) Strengths:**

First, this paper provides very comprehensive experimental evidence that clearly demonstrates the advantages of the proposed method in retrieval tasks and extrapolation capability.

**(2) Weaknesses:**

This paper essentially proposes a method that combines sparse attention with linear models. There already exist numerous similar approaches in the literature, and a thorough investigation and discussion of these related methods would be necessary. Of course, if the authors were unaware of certain related methods, that would be understandable, but the baseline comparisons in the paper need to be more complete and fair. For instance, the chunk selection process for the NLA portion resembles existing efficient sparse attention methods such as NSA or MoBA; therefore, comparisons should not be limited to linear models that are not particularly competitive in certain capabilities. Additionally, the paper does not specify the chunk sizes and quantities used for the NLA and LA portions. While it can be inferred that all selected chunks in the LA portion can be merged into a fixed-size state, the number and size of chunks used in the NLA portion remain unclear. Furthermore, the selection process and the merging process for selected chunks are not described in detail, leading to significant confusion—only the final merged result using Equation 20 is presented. The computational complexity of the selection or routing mechanism needs to be discussed, ideally as part of the overall complexity analysis, so that the end-to-end computational complexity of the entire process can be provided, rather than reporting only the complexity of the actual NLA and LA attention computations while excluding the routing overhead. Based on such complexity analysis, the method should be sublinear; therefore, an explanation is needed for why the decoding time for NAtS-L Hybrid in Figure 4 remains almost constant. Is this because the routing process (Equation 8) is omitted during decoding, and the model simply uses the previously determined NLA and LA portions directly?

---

> ### Author Rebuttal · Authors · 2026-03-31
>
> Dear reviewer M6Vf,
>
>  we thank you for your constructive feedback. We also appreciate that you highlight NAtSL's strengths in retrieval and extrapolation tasks.
>
> Regarding your concerns:
> > There already exist numerous similar approaches in the literature,
>
> We briefly discussed them in section 2 (lines 114-129) and showed the difference between NAtS-L and the other hybrid models. Thanks for pointing out that they should be discussed in more detail. We will adapt this in our manuscript.
>
> > Comparison between NSA and MOBA.
>
> The result is presented under https://drive.google.com/file/d/1jNmk-qFykNFj818FisCace5g2GhA3pdA/view?usp=sharing , Table 1. MOBA and NSA perform similarly to the full transformer models, while NAtS-L could achieve a higher retrieval score.
>
>
> However, we would still argue that NAtS-L is not a simple incremental work towards these two approaches. We agree that both approaches implemented the chunk selection process.  However, each Q vector in both NSA and MOBA can still only compute the interaction with a fraction (the top-k chunks) of KV matrices and ignore all the other correlations. NAtS-L instead builds a correlation between Q vectors and the other non-softmax attention chunks with linear attention.  Additionally, NAtS-L does not enforce top-k selection within each layer but learns to assign tokens to either softmax or linear attention chunks via gradient descent. Therefore, NAtS-L provides more flexible budget distributions over different layers. Last but not least, both NSA and MOBA still maintain the full KV cache, while NAtS-L can efficiently compress the linear attention part to a single hidden state. Hence,  NAtS-L is not orthogonal to NSA and MOBA. Instead, applying MOBA and NSA to NAtS-L’s softmax attention chunks could further reduce NAtS-L's computational costs.
>
> > the chunk sizes are not specified
>
> We thank the reviewer for pointing out the uncertainty and will adapt the manuscript accordingly. We briefly discuss the chunk size in lines 243-246, stating that the NAtS-L chunk size should be no smaller than the GDN chunk size to have a better trade-off between model performance and training efficiency. Additionally, we want the chunk size to be as small as possible, while remaining flexible enough to work with different contexts. Hence, following the GDN implementation, we use a chunk size of 64 for NAtS-L. We also have an ablation study in Table 2 of the linked files.
>
>  > Quantities used for the NLA and LA portions.
>
> Everything is learned end-to-end with gradient descent, so we do not specifically control the fraction of LA and NLA, the operation applied to each chunk only depends on the scores for the corresponding chunks. Hence, different layers (and even the same layer for different input contexts) could have different LA-NLA fractions. We illustrate the fraction in Section 5.2, Figures 5 and 8. We show that some heads only applied LA porion, some mostly with NLA portions, while the other heads provide mixed attention patterns.
>
> > Computational complexity analysis
>
> The routing operation is shown in Equation 8. We score each operation (LA and NLA) independently using the feature map values within each chunk. For each chunk, the operation with the highest score will be selected, regardless of the scores in the other chunks, and we do not enforce a top-k selection across chunks. Hence, the routing operation has constant computational complexity.  During the decoding stage, the decoding time is mainly bounded by the memory I/O process. Therefore, NAtS-L does not increase as much as Transformers, as interacting with memory remains the main bottleneck at this sequence length level.
>
> > Include fairer experimental comparisons
>
> Besides MOBA and NSA, we also compared against GDN + sliding window attention with a window size of 1024. Surprisingly, this variation performs poorly on the given tasks. This might be the result of overemphasizing the last few tokens while ignoring the previous ones. This highlights the importance of the adaptive token selection process.
>
> > There are many methods that combine sparse attention with linear attention.
>
>  To the best of our knowledge, we are the first to propose a token-level end-to-end adaptive hybrid model that learns to select the optimal operation from scratch. Other approaches either interleave only between LA and NLA layers, require a pre-trained transformer that can be approximated, or follow a fixed rule with hand-crafted hyperparameters. However, given that each token may contain different amounts of information in different contexts, a fixed rule might not capture the semantic information in each token. Hence, we propose an adaptive token-level hybrid approach that better handles different types of context information.
>
> > Regarding efficient implementation
>
>  To avoid unnecessary computation, many operations are rewritten from scratch using Triton, but the main workflow follows the implementation in FLA and FlashAttention.

---

> > ### Author Rebuttal · Reviewer_M6Vf · 2026-04-01
> >
> > For more details, see my comment

---

> > > ### Author Response · Authors · 2026-04-01
> > >
> > > Dear Reviewer M6Vf,
> > >
> > > We are glad that we have addressed your concerns, and thank you for raising your score!

---

### Decision · Program_Chairs · 2026-04-30

**Decision:**

Accept (regular)

**Comment:**

This paper introduces Neural Attention Search Linear (NATS-L), a framework that dynamically routes tokens to either linear or softmax attention within the same layer to achieve a strong trade-off between efficiency and quality.

The reviewers uniformly recommended a weak accept, agreeing that the paper proposes a well-motivated and technically solid method. The experimental results clearly demonstrate the advantages of NATS-L in retrieval tasks and extrapolation capability across various benchmarks. During the rebuttal phase, the authors successfully resolved initial concerns regarding baseline comparisons, missing hyperparameter details, and the empirical verification of the hybrid balance. While reviewers noted the work is an extension of existing hybrid architectures rather than a fundamental theoretical advance, they agreed it makes a sensible contribution to efficient long-context modeling. Therefore, I recommend the paper for acceptance to the conference.